# REGRESSING THE RELATIVE FUTURE:
# EFFICIENT POLICY OPTIMIZATION FOR MULTI-TURN RLHF

**Zhaolin Gao[1], Wenhao Zhan[2], Jonathan D. Chang[3]\*, Gokul Swamy[4],**
**Kianté Brantley[5], Jason D. Lee[2], Wen Sun[1]**
[1] Cornell University, [2] Princeton University, [3] Databricks Mosaic Research,
[4] Carnegie Mellon University, [5] Harvard University

## ABSTRACT

Large Language Models (LLMs) have achieved remarkable success at tasks like summarization that involve a *single turn* of interaction. However, they can still struggle with *multi-turn* tasks like dialogue that require long-term planning. Previous works on multi-turn dialogue extend single-turn reinforcement learning from human feedback (RLHF) methods to the multi-turn setting by treating all prior dialogue turns as a long context. Such approaches suffer from *covariate shift*: the conversations in the training set have previous turns generated by some reference policy, which means that low training error may not necessarily correspond to good performance when the learner is actually in the conversation loop. In response, we introduce REgressing the RELative FUture (REFUEL), an efficient policy optimization approach designed to address multi-turn RLHF in LLMs. REFUEL employs a single model to estimate $Q$-values and trains on self-generated data, addressing the covariate shift issue. REFUEL frames the multi-turn RLHF problem as a sequence of regression tasks on iteratively collected datasets, enabling ease of implementation. Theoretically, we prove that REFUEL can match the performance of any policy covered by the training set. Empirically, we evaluate our algorithm by using Llama-3.1-70B-it to simulate a user in conversation with our model. REFUEL consistently outperforms state-of-the-art methods such as DPO and REBEL across various settings. Furthermore, despite having only 8 billion parameters, Llama-3-8B-it fine-tuned with REFUEL outperforms Llama-3.1-70B-it on long multi-turn dialogues. Implementation of REFUEL can be found at https://github.com/ZhaolinGao/REFUEL/, and models trained by REFUEL can be found at https://huggingface.co/Cornell-AGI.

## 1 INTRODUCTION

Despite the impressive performance Large Language Models (LLMs) have demonstrated on tasks like summarization, question answering, and short conversations (OpenAI, 2023; Meta, 2024; Google, 2024; Anthropic, 2024), most LLMs struggle with *planning* effectively for long conversations that involve multiple rounds of dialogue or asking follow-up questions about previous responses (Irvine et al., 2023; Abdulhai et al., 2023). The root cause of this deficiency is that most preference fine-tuning methods using Reinforcement Learning from Human Feedback (RLHF, Christiano et al. (2017); Ziegler et al. (2020); Ouyang et al. (2022); Rafailov et al. (2024b); Azar et al. (2023); Guo et al. (2024); Rosset et al. (2024); Dong et al. (2024); Gao et al. (2024); Wu et al. (2024); Meng et al. (2024)) treat all tasks as *single-turn* (i.e. as a contextual bandit (Auer et al., 2002; Langford & Zhang, 2007)) even when some tasks are fundamentally multi-turn (e.g. a multi-step dialogue with a user).

The simplest way to convert a multi-turn task, such as dialogue, into a single-turn task is to train on the last-turn of the dialogue and use dialogue history as context. Although this approach is appealing due to its

---

\*Work done at Cornell

{zg292,ws455}@cornell.edu, wenhao.zhan@princeton.edu, j.chang@databricks.com, gswamy@andrew.cmu.edu, kdbrantley@g.harvard.edu, jasonlee@princeton.edu

Figure 1: We present REFUEL: a simple, regression based approach for multi-turn RLHF. Traditional single-turn RLHF methods suffer from *covariate shift* as they train on histories generated by the *base* policy rather than *current* policy. REFUEL eliminates the covariate shift by iteratively generate on-policy datasets, aligning the training and testing distributions. REFUEL performs better at later turns compared to the baseline methods in terms of winrate (which is computed against the base policy, Llama-3-8B-it, using GPT4).

compatibility with pre-existing pipelines, training on histories generated by the base policy rather than current policy introduces covariate shift (Kohavi, 1995) between the training and testing distributions. This can result in poor performance when the learner is in the conversation loop (Ross et al., 2011). This phenomenon is at the heart of recent empirical observations by Zhou et al. (2024b) about LLMs struggling to ask questions to clarify missing information in conversations or being unable to self-correct after making mistakes in mathematical reasoning tasks, as shown in the concurrent work of Kumar et al. (2024a).

In response, several authors have proposed treating multi-turn tasks like dialogue as proper online RL problems, rather than contextual bandits. For example, Zhou et al. (2024a); Shani et al. (2024a) propose applying an online actor-critic framework to allow the policy to learn to respond to its own past decisions (e.g. asking for more information or correcting mistakes), improving practical performance. However, actor-critic methods substantially increase the training complexity (in terms of both stability and memory usage), especially when both the actor and the critic are LLMs with billions of parameters.

To handle policy-induced covariate shift without necessitating an extra critic network, we propose using the reparameterization trick introduced by Degrave et al. (2019); Rafailov et al. (2024b) to regress future returns (i.e. $Q$-values) in terms of the log of policy ratios, allowing us to "read-off" the corresponding soft optimal policy, replacing the usual two-step procedure of fitting a critic followed by an explicit policy optimization step with a single step procedure. However, it is not clear how to get supervision for this regression problem, as learned reward models can only provide accurate feedback at the *trajectory* (e.g. conversation) level rather than at the *turn* (e.g. generation) level, which means we cannot apply techniques for learning a $Q$ function that require per-timestep reward labels. Our key insight is that the difference in two conversation-level rewards generated from a shared prefix is an unbiased estimate of the difference in $Q$ values between the first divergent turns. We can then use this difference in $Q$-values as a regression target to adapt *any* pair-wise preference-based single-turn RLHF method to the multi-turn setting without introducing *any* additional components. Crucially, we iteratively generate on-policy datasets of prefixes and two independent completions from the current policy. Training on this on-policy data ensures that the model learns to participate in the sort of conversations it would actually encounter when participating in the conversational loop, rather than those in some offline dataset, addressing the covariate shift issue that stymies offline single-turn methods. We call this approach REFUEL: **RE**gressing the **RE**lative **FU**tur**E** for reinforcement **L**earning.

## 2 PRELIMINARIES

Consider a conversation between a human and an AI assistant. Let the initial question from the human be denoted as $x_1 \sim \rho$. Upon receiving $x_1$, the AI assistant, $\pi$, generates a response $y_1 \sim \pi(\cdot|x_1)$. Subsequently, given $x_1, y_1$, the human responds in turn $x_2 \sim T(\cdot|x_1, y_1)$, where $T(\cdot|\{x_i, y_i\}_{i=1}^n)$ denotes the conditional distribution of the human responses $x_{h+1}$. Upon receiving $x_2$, the AI assistant generates a second response

$y_2 \sim \pi(\cdot|x_1, y_1, x_2)$. This interactive process continues until we reach the total number of turns $H$. At the end of the interaction, the AI assistant receives a trajectory-level reward $r(\{x_h, y_h\}_{h=1}^H)$. In this work, we do not focus on the learning process of the reward function $r$; instead, we utilize existing pre-trained reward models.

We can cast the multi-turn RLHF as a standard multi-step Markov Decision Process (MDP) by using the conversational transcript as state. Let the *state* $s_h$ at turn $h$ comprise all prior information up to turn $h$, excluding the current response: $s_h = \{x_1, y_1, \ldots, x_{h-1}, y_{h-1}, x_h\}$. Then, the response $y$ can be interpreted as an *action*. We denote the state and action spaces at step $h$ as $\mathcal{S}_h$ and $\mathcal{Y}_h$ respectively. For simplicity, we assume $|\mathcal{Y}_h| = Y$ for all $h \in [H]$. The policy $\pi$ maps from a state $s_h$ to the next response $y_h$, i.e., $y_h \sim \pi(\cdot|s_h)$. We denote $d_h^\pi(s)$ as the state distribution at turn $h$ induced by the policy $\pi$, with $s_h \sim d_h^\pi$ as the process of sampling $s_h$ from $\pi$. The policy receives a reward $r(s_{H+1})$ after step $H$ where, for notation convenience, we denote $s_{H+1} = (s_H, y_H)$. Note that $s_{H+1}$ is the entire multi-turn conversation. The dynamics $P(s_{h+1}|s_h, y_h)$ are fully determined by $T$ that governs the response generation process of the human, i.e., $x_{h+1} \sim T(\cdot|s_h, y_h)$ and $s_{h+1} := \{s_h, y_h, x_{h+1}\}$. We emphasize that in contrast to the standard single-turn RLHF setting which is often modeled by a deterministic MDP or bandit problem, the transition $P$ is random as $T$ is random.

**Rollins & Rollouts.** Given a state $s_h$ and a response $y_h$, we denote by $s_{H+1} \hookleftarrow \pi(s_h, y_h)$ the process of sampling the final state by generating response $y_h$ at $s_h$ followed by executing $\pi$ until turn $H$ (i.e., finishing the entire conversation). We refer to this process as a *rollout* of policy $\pi$. Following the standard RL notation, we denote $Q_h^\pi(s_h, y_h)$ as the state-action Q function which models the expected future reward-to-go of the random process of taking $y_h$ at $s_h$ followed by rolling out $\pi$ to the end. Similarly, given a turn step $h$, we use *rollin* to refer to the process of sampling a state at turn $h$, denoted as $s_h \sim d_h^\pi$.

**Resets.** Given a state $s_h$, *resetting to $s_h$* simply means that the policy $\pi$ starts from $s_h$ again and generates counter-factual trajectories from $s_h$. While resets are often considered as a strong assumption in general RL, it is *trivially achievable* in the context of RLHF for text generation. Resetting to $s_h$ can be implemented by feeding the partial conversation $s_h = \{x_1, y_1, \ldots, x_h\}$ to the transformer-based policy $\pi$ as a prefix / context. This capability allows a policy to generate multiple independent future trajectories from the same state $s_h$.

## 2.1 The Limitation of Single-turn RLHF Methods on Multi-turn Problems

Recent RLHF algorithms such as DPO (Rafailov et al., 2024b), IPO (Azar et al., 2023), SPPO (Wu et al., 2024), and REBEL (Gao et al., 2024) are specifically designed for the single-turn setting which can be formulated as a contextual bandit problem with $H = 1$. When applying these methods to multi-turn datasets such as Anthropic HH (Bai et al., 2022), it is common to first convert from multi-turn into a single-turn format. Specifically, for each sequence of multi-turn interactions $\{x_1, y_1, x_2, y_2, \ldots, x_H, y_H\}$, these single-turn methods treat the first $H-1$ interactions as a large context $x := \{x_1, y_1, \ldots, x_H\}$, and only optimize the last-turn generation of $y_H$. Consequently, the dataset consists of $\{x \sim \mathcal{D}_{\text{off}}, y \sim \pi(\cdot|x), y' \sim \pi(\cdot|x)\}$ where we use $\mathcal{D}_{\text{off}}$ denotes the offline dataset. This approach is used by Rafailov et al. (2024b) to optimize the multi-turn Anthropic HH dataset.

As depicted in Figure 1, applying single-turn RLHF methods to a multi-turn setting in this manner introduces *covariate shift* (Kohavi, 1995) between training and testing distributions. Intuitively, the resulting policy has only learned to generate the final response based on the contexts present in the offline data. However, during inference, the policy is likely to observe different contexts, as they are generated by itself, rather than the policy used to collect the offline dataset. This can lead to degraded performance at test time, paralleling the issues with offline approaches to imitation learning like behavioral cloning first formalized by Ross et al. (2011).

## 3 REFUEL: REgressing the RELative FUture

To address covariate shift in multi-turn RLHF without introducing the overhead of an explicit critic network, we introduce REFUEL. REFUEL eliminates the need of an explicit critic by merging the two-step process of actor-critic algorithms into a unified procedure and reduces covariate shift by using on-policy datasets. At each iteration $t$, REFUEL aims to solve the following KL-constrained RL problem:

$$\pi_{t+1} = \underset{\pi \in \Pi}{\arg\max} \, \mathbb{E}_{h, s_h, y_h \sim \pi_t(\cdot|s_h)} Q_h^{\pi_t}(s_h, y_h) - \frac{1}{\eta} \mathbb{E}_{h, s_h} \text{KL}(\pi(\cdot|s_h) \| \pi_t(\cdot|s_h)) \tag{2}$$

---

**Algorithm 1** REgressing the RElative FUturE for reinforcement Learning (REFUEL)

---

**Require:** number of iterations $T$, learning rate $\eta$, trajectory-level reward model $r(\cdot)$.
1: Initialize policy $\pi_1$.
2: **for** $t = 1 \ldots T$ **do**
3:   Collect dataset $\mathcal{D} = \{h, s_h, y_h, y'_h, s_{H+1}, s'_{H+1}\}$ where

$$h \sim U(H), s_h \sim d_h^{\pi_t}, y_h \sim \pi_t(\cdot|s_h), y'_h \sim \pi_t(\cdot|s_h), s_{H+1} \twoheadleftarrow \pi_t(s_h, y_h), s'_{H+1} \twoheadleftarrow \pi_t(s_h, y'_h)$$

4:   Update policy via regression to relative future rewards:

$$\pi_{t+1} = \underset{\pi}{\arg\min}\, \widehat{\mathbb{E}}_{\mathcal{D}} \left( \frac{1}{\eta} \left( \ln \frac{\pi(y_h|s_h)}{\pi_t(y_h|s_h)} - \ln \frac{\pi(y'_h|s_h)}{\pi_t(y'_h|s_h)} \right) - \underbrace{(r(s_{H+1}) - r(s'_{H+1}))}_{\text{Relative Future Reward}} \right)^2 \quad (1)$$

5: **end for**

---

Intuitively, the policy $\pi_{t+1}$ is chosen to maximize the expected reward (through $Q$-values) while simultaneously minimizing the change from the previous policy $\pi_t$, with the balance determined by parameter $\eta$. From Ziebart et al. (2008), we know there exists a closed-form solution to the above minimum relative entropy problem:

$$\forall h, s_h, y_h : \pi_{t+1}(y_h|s_h) = \frac{\pi_t(y_h|s_h) \exp(\eta Q_h^{\pi_t}(s_h, y_h))}{Z(s_h)}; Z(s_h) = \sum_{y_h} \pi_t(y_h|s_h) \exp(\eta Q_h^{\pi_t}(s_h, y_h)) \quad (3)$$

Following Degrave et al. (2019); Rafailov et al. (2024b), we can rearrange Eq. 3 to express the $Q$-value as a function of the policy:

$$\forall h, s_h, y_h : Q_h^{\pi_t}(s_h, y_h) = \frac{1}{\eta} \left( \ln Z(s_h) + \ln \frac{\pi_{t+1}(y_h|s_h)}{\pi_t(y_h|s_h)} \right). \quad (4)$$

Note that the partition function $Z(s_h)$ does not depend on $y_h$ and that we can sample another response $y'_h$ by resetting $\pi_t$ to $s_h$, $y'_h \sim \pi_t(\cdot|s_h)$. By taking the difference of the above expression across the paired responses $(y_h, y'_h)$ we can eliminate the partition function:

$$\forall h, s_h, y_h, y'_h : Q_h^{\pi_t}(s_h, y_h) - Q_h^{\pi_t}(s_h, y'_h) = \frac{1}{\eta} \left( \ln \frac{\pi_{t+1}(y_h|s_h)}{\pi_t(y_h|s_h)} - \ln \frac{\pi_{t+1}(y'_h|s_h)}{\pi_t(y'_h|s_h)} \right). \quad (5)$$

Following Gao et al. (2024), we can then formulate satisfying the above constraint as a least squares problem:

$$\left( \frac{1}{\eta} \left( \ln \frac{\pi_{t+1}(y_h|s_h)}{\pi_t(y_h|s_h)} - \ln \frac{\pi_{t+1}(y'_h|s_h)}{\pi_t(y'_h|s_h)} \right) - \left( Q_h^{\pi_t}(s_h, y_h) - Q_h^{\pi_t}(s_h, y'_h) \right) \right)^2 \quad (6)$$

Unfortunately, this loss function uses $Q$-values, which we do not have direct access to. However, the reward obtained from a rollout starting from $s_h$ is an unbiased estimate of the $Q$-value. We perform independent policy rollouts using $\pi_t$ at $(s_h, y_h)$ and $(s_h, y'_h)$, obtaining the ending states $s_{H+1}$ and $s'_{H+1}$ from the two independent rollouts (i.e., $s_{H+1} \twoheadleftarrow \pi_t(s_h, y_h)$, $s'_{H+1} \twoheadleftarrow \pi_t(s_h, y'_h)$). The rewards of these states $r(s_{H+1})$ and $r(s'_{h+1})$ have expected values of $Q_h^{\pi_t}(s_h, y_h)$ and $Q_h^{\pi_t}(s_h, y'_h)$. Then, leveraging the fact that the minimizer of a least squares problem is the conditional mean of the target variable, we arrive at Eq. 1. The pseudocode of our algorithm is provided in Alg. 1 where $\widehat{\mathbb{E}}_{\mathcal{D}}$ denotes the empirical average over the dataset $\mathcal{D}$. To reduce the computational complexity of REFUEL, we uniformly sample a turn step $h$ during training similar to Ross & Bagnell (2014). Notably, the use of pairwise rollouts with a shared prefix and unbiased estimates of $Q$-values enables REFUEL to leverage any single-turn RLHF optimizer. Here, we follow the derivations of REBEL, leading to the same update formulation as REBEL.

## 3.1 Intuitive Explanation of REFUEL

From our above argument, we know that solving Equation 1 optimally would imply that

$$\forall h, s_h, y_h, y'_h : \frac{1}{\eta}\left(\ln\frac{\pi_{t+1}(y_h|s_h)}{\pi_t(y_h|s_h)} - \ln\frac{\pi_{t+1}(y'_h|s_h)}{\pi_t(y'_h|s_h)}\right) = Q_h^{\pi_t}(s_h, y_h) - Q_h^{\pi_t}(s_h, y'_h).$$

Summing the above over $y'_h$ further implies that there must exist a $y$-independent function $c_h(s_h)$ such that

$$\forall h, s_h : \frac{1}{\eta}\ln\frac{\pi_{t+1}(y_h|s_h)}{\pi_t(y_h|s_h)} = Q_h^{\pi_t}(s_h, y_h) - c_h(s_h).$$

Rearranging the terms, we can write that

$$\forall h, s_h, y_h : \pi_{t+1}(y_h|s_h) = \pi_t(y_h|s_h)\exp\left(\eta Q_h^{\pi_t}(s_h, y_h) - \eta c_h(s_h)\right) \propto \pi_t(y_h|s_h)\exp\left(\eta Q_h^{\pi_t}(s_h, y_h)\right).$$

Note that that $\eta c_h(s_h) = \ln\mathbb{E}_{y\sim\pi_t(.|s_h)}\exp(\eta Q_h^{\pi_t}(s_h, y_h)) = Z(s_h)$ is the log-partition function. In our algorithm REFUEL, we predict the relative future rewards instead of modeling the partition function using an additional critic network. Prior works do not leverage the idea of predicting relative values: they either assume that the partition function is approximately equal to a constant (Zhu et al., 2023) or use an another critic function to approximate it, incurring extra GPU memory and computation costs (Wu et al., 2024; Richemond et al., 2024). We also note that that above policy update procedure recovers the NPG update with the softmax policy parametrization (Agarwal et al., 2021), which converges to the globally optimal policy at the rate of $O(1/T)$, a faster rate compared to that of standard policy gradient methods.

## 3.2 More rigorous analysis and connection to past policy gradient theory

The above simplified explanation relies on an unrealistic assumption that least square regression can learns the Bayes optimal predictor exactly in a point-wise manner. In this section, we analyze the performance of REFUEL under a much more realistic assumption — we assume that the learned predictor in Equation 1 can predict well on average under the training distribution. Our analysis below extends that of REBEL (Gao et al., 2024) from the bandit setting to multi-turn MDPs with stochastic transitions. We denote $\mathcal{S}_h$ as the set of all possible states at time step $h$, and we assume $\mathcal{S}_h$ and $\mathcal{S}_{h'}$ for $h \neq h'$ are disjoint. This assumption is satisfied in the multi-turn RLHF setting since $s_h$ and $s_{h'}$ model states with different numbers of turns. We start by assuming the learned predictor from the least square regression problem in Equation 1 has bounded in-distribution generalization error.

**Assumption 1.** *There exists an $\epsilon \in \mathbb{R}^+$, such that for all $t$,*

$$\mathbb{E}_{h,s_h\sim d_h^{\pi_t}, y_h\sim\pi_t(\cdot|s_h), y'_h\sim\pi_t(\cdot|s_h)}\left(\frac{1}{\eta}\left(\ln\frac{\pi_{t+1}(y_h|s_h)}{\pi_t(y_h|s_h)} - \ln\frac{\pi_{t+1}(y'_h|s_h)}{\pi_t(y'_h|s_h)}\right) - \left(Q_h^{\pi_t}(s_h, y_h) - Q_h^{\pi_t}(s_h, y'_h)\right)\right)^2 \leq \epsilon.$$

In the above assumption, we have bounded prediction error to the Bayes optimal, the relative Q value – $\left(Q_h^{\pi_t}(s_h, y_h) - Q_h^{\pi_t}(s_h, y'_h)\right)$, under the online data distribution. For regret bound, we will compare to a policy that is covered by the training distributions.

**Assumption 2** (Coverage). *We say that a comparator policy $\pi^*$ (not necessarily a global optimal policy) is covered by the training distributions, if the following two concentrability coefficients are bounded for all $t$:*

$$C_{s;\pi^*} := \max_{h,s_h,y_h,t}\frac{d_h^{\pi^*}(s_h)}{d_h^{\pi_t}(s_h)} < \infty, \quad C_{y;\pi^*} := \max_{h,s_h,y_h,t}\frac{\pi^*(y_h|s_h)}{\pi_t(y_h|s_h)} < \infty.$$

The first concentration coefficient $C_{s;\pi^*}$ concerns the state distribution, while the second one concerns the coverage in the single-turn response (i.e., action) space. These concentrability coefficients play key role in policy gradient theorem (e.g., Kakade & Langford (2002); Bagnell et al. (2003); Abbasi-Yadkori et al. (2019); Agarwal et al. (2021); Bhandari & Russo (2024)). In our definition, we use iteration-dependent on-policy distributions $d^{\pi_t}$ and $\pi_t$ to capture the case where on-policy distributions happen to be informative in terms of covering a good comparator policy (e.g., initialization $\pi_0$ – typically is a pre-trained LLM, is informative

in terms of covering a high quality policy). While we focus on pure on-policy algorithm, similar to REBEL, incorporating additional offline distribution into the algorithm and analysis is straightforward.

Denote $J(\pi)$ as the expected total reward of the policy $\pi$. REFUEL has the following performance guarantee.

**Theorem 1.** *Under Assumption 1 and Assumption 2, if we initialize $\pi_1$ to be a uniformly random policy and choose an appropriate $\eta$, after $T$ iterations, there must exist a policy $\pi_t$ where $t \in [T]$ such that for all comparator policy $\pi^*$,*

$$J(\pi^*) - J(\pi_t) \le O\left(H\sqrt{\frac{1}{T}} + H\sqrt{C_{s;\pi^*}C_{y;\pi^*}\epsilon}\right).$$

The above theorem indicates that as long as least square regressions are successful, i.e., in-distribution generalization error $\epsilon$ is small, we can learn at least as well as any policy $\pi^*$ covered by the training data. Note that, in general, when learning is involved, we should not expect to compete against the globally optimal policy since PG methods cannot do strategic exploration. We now discuss the situation where Assumption 1 holds by connecting and comparing it to similar conditions used in prior policy gradient theory.

**Discussion on Assumption 1.** One condition where $\epsilon$ in Assumption 1 can be small is the *Approximate Policy Completeness (APC)* condition.

**Definition 1.** *There exists $\epsilon_\Pi \in \mathbb{R}^+$, such that for all $\pi \in \Pi$,*

$$\min_{C \in \mathcal{S} \mapsto \mathbb{R}^+} \min_{\pi' \in \Pi} \mathbb{E}_{h, s_h \sim d_h^\pi, y_h \sim \pi}\left(\frac{1}{\eta}\ln \pi'(y_h|s_h) - \frac{1}{\eta}\ln \frac{\pi(y_h|s_h)\exp(\eta Q_h^\pi(s_h, y_h))}{C(s_h)}\right)^2 \le \epsilon_\Pi,$$

Note that $C(s)$ can be **any function from $\mathcal{S} \to \mathbb{R}^+$** as long as it is independent of the $y$. To get a better understanding of the above assumption, let us first show that the following *soft policy improvement closure property* implies the above condition. The soft policy improvement closure condition means that for all $\pi \in \Pi$, we have $\pi(y|s)\exp(\eta Q_h^\pi(s, y))/Z(s) \in \Pi$ (here $Z$ is the partition function). In this case, we set $C(s) := Z(s)$, and the soft improvement policy $\pi'(y|s) := \pi(y|s)\exp(\eta Q_h^\pi(s, y))/Z(s)$ is the minimizer and we have $\epsilon_\Pi = 0$. On the other hand, in general, we note that $C$ does not have to be equal to the partition function $Z$. In fact, our condition allows us to select the $C$ that delivers the *smallest* APC error $\epsilon_\Pi$. This is possible in our case since our algorithm is performing regression to *relative* Q values. Conditions similar to APC are common sufficient conditions for the success of policy optimization methods (e.g., CPI (Kakade & Langford, 2002), PSDP (Bagnell et al., 2003), PG (Bhandari & Russo, 2024), and NPG (Agarwal et al., 2021))[1]. While it is not a necessary condition, it is known that the standard realizability condition alone (i.e., just assume $\pi^* \in \Pi$) is not sufficient for permitting efficient policy learning in general (Jia et al., 2024).

Now we show that by using pairs and performing regression to future reward difference, our APC condition is strictly weaker than the conditions required in previous NPG analysis (Agarwal et al., 2021). This is formalized in the following example using log-linear polices.

**Remark 1.** *The APC condition is strictly weaker than the small Q function approximation error assumption in NPG methods (Agarwal et al., 2021) in the linear setting. In particular, we have the following propositions.*

**Proposition 1.** *Given a feature mapping $\phi : \mathcal{S} \times \mathcal{Y} \mapsto \mathbb{R}^d$, let $\Pi$ denote the log-linear policy class:*

$$\Pi = \left\{\pi : \exists \theta \in \mathbb{R}^d, \forall (s_h, y_h), \pi(y_h|s_h) \propto \exp\left(\theta^\top \phi(s_h, y_h)\right)\right\}.$$

*If we can bound the Q function approximation error using linear function on $\phi$ — a key condition Agarwal et al. (2021) use for proving NPG convergence:*

$$\forall \pi \in \Pi : \min_{w \in \mathbb{R}^d} \mathbb{E}_{h, s_h \sim d_h^\pi, y_h \sim \pi}\left[\left(Q_h^\pi(s_h, y_h) - w^\top \phi(s_h, y_h)\right)^2\right] \le \epsilon, \tag{7}$$

*then the APC condition is satisfied with $\epsilon_\Pi \le \epsilon$.*

---

[1]Prior methods typically require APC under a *hard policy improvement procedure*. The simplified version of their conditions can be intuitively understood as $\text{argmax}_a Q_h^\pi(s, a) \in \Pi$ for all $\pi \in \Pi$, which corresponds to $\eta = \infty$ in the soft policy improvement closure.

*Conversely, there exists an instance where the APC condition is satisfied with $\epsilon_\Pi = 0$ while the Q function approximation error can be as large as* 1:

**Proposition 2.** *Consider the same log-linear policy class. There exists an MDP, feautre mapping $\phi$ and $\pi \in \Pi$ such that the APC condition is satisfied with $\epsilon_\Pi = 0$ but*

$$\min_{w \in \mathbb{R}^d} \mathbb{E}_{h, s_h \sim d_h^\pi, y_h \sim \pi} \left[ \left( Q_h^\pi(s_h, y_h) - w^\top \phi(s_h, y_h) \right)^2 \right] = 1.$$

*These two propositions formally demonstrate that APC is weaker than the condition required for proving NPG convergence, showing the theoretical benefit of using pairs of rollouts and regressing to relative futures.*

Further discussions on the above propositions can be found in Appendix C, and the variance reduction effect of REFUEL is discussed in Appendix D.

## 4 EXPERIMENTS

Our implementation closely follows the pseudocode in Alg. 1. We empirically evaluate REFUEL's ability under two multi-turn RLHF settings. In the first setting, we create a multi-turn conversation simulator that uses Llama-3.1-70B-it to simulate a human-in-the-loop. In the second setting, we evaluate our approach using a pre-sampled sequence of questions from existing multi-turn RLHF datasets to simulate multi-turn dialogue. The first setting models a realistic situation where the learning agent and the user need to interact, while the second setting models a simplified situation where the sequence of human questions is pre-sampled before the conversation begins. However, even in the second setting, the learning agent still needs to learn to generate future turns conditioned on its own previous turns. Additional experiment details are in Appendix F.

### 4.1 BASELINES: SINGLE-TURN AND MULTI-TURN

We compare REFUEL to single-turn and multi-turn baselines that are extensions of three algorithms, RLOO (Kool et al., 2019), DPO (Rafailov et al., 2024b) and REBEL Gao et al. (2024), as well as two open-source LLMs: Llama-3.1-8B-it and Llama-3.1-70B-it (Meta, 2024). For the single-turn baselines, we consider the following three settings:

**Last-Turn-Offline** (LT-OFFLINE): This is a standard approach to applying single turn methods to a multi-turn RLHF dataset. Specifically, we rollin using offline data and train the last turn on pairs of offline responses, $\mathcal{D} = \{(s_H, y_H, y_H') \sim \mathcal{D}_{\text{off}}, s_{H+1} = (s_H, y_H), s_{H+1}' = (s_H, y_H')\}$. For RLOO and REBEL, the rewards are computed using $s_{H+1}$ and $s_{H+1}'$, while DPO selects chosen and rejected responses based on the reward values.

**Last-Turn-Mixed** (LT-MIXED): This is another standard approach, similar to LT-OFFLINE, where we rollin using the offline data. However, on the last turn, we sample and train on pairs of on-policy rollouts responses: $\mathcal{D} = \{s_H \sim \mathcal{D}_{\text{off}}, y_H \sim \pi_t(\cdot|s_H), y_H' \sim \pi_t(\cdot|s_H), s_{H+1} = (s_H, y_H), s_{H+1}' = (s_H, y_H')\}$.

**Last-Turn-Online** (LT-ONLINE): Unlike the previous two approaches, this approach involves using on-policy samples rather offline data for both the rollin and rollout responses. Specifically, the state $s_H$ and the responses are generated from the current policy with a simulated user, denoted as $\mathcal{D} = \{s_H \sim d_H^{\pi_t}, y_H \sim \pi_t(\cdot|s_H), y_H' \sim \pi_t(\cdot|s_H), s_{H+1} = (s_H, y_H), s_{H+1}' = (s_H, y_H')\}$.

For the three single-turn baselines mentioned previously, we always rollin and optimize the last turn $H$. In multi-turn baseline approaches, we rollin and optimize each turn instead of only optimizing at the last turn. We consider one multi-turn baseline approach similar to the baseline proposed in Shani et al. (2024a):

**Multi-Turn-Mixed** (MT-MIXED): Similar to the LT-MIXED approach, we rollin with the offline data, but now we sample on-policy pair of responses at an arbitrary state $s_h$ from the offline dataset. After sampling a state, we perform two rollouts from $s_h$ to the end $H$: $\mathcal{D} = \{h \sim U(H), s_h \sim \mathcal{D}_{\text{off}}, y_h \sim \pi_t(\cdot|s_h), y_h' \sim \pi_t(\cdot|s_h), s_{H+1} \curvearrowleft \pi_t(s_h, y_h), s_{H+1}' \curvearrowleft \pi_t(s_h, y_h')\}$. The rewards are computed using $s_{H+1}$ and $s_{H+1}'$, which are the unbiased estimates of the $Q$-values at turn $h$. This baseline optimizes future returns similar to REFUEL, but at the states sampled from the offline data.

The detailed dataset statistics for each method are provided in Appendix F.4. We chose not to compare against PPO baselines (Shani et al., 2024a; Zhou et al., 2024b) due to its computational inefficiency. Training with

| Method | Winrate at Turn (↑) | | | | | avg |
|---|---|---|---|---|---|---|
| | $h = 1$ | $h = 2$ | $h = 3$ | $h = 4$ | $H = 5$ | |
| Llama-3.1-8B-it | 57.8 | 57.8 | 52.4 | 55.2 | 54.0 | 55.44 |
| Llama-3.1-70B-it | 70.4 | 66.4 | 61.0 | 53.0 | 55.4 | 61.24 |
| RLOO-LT-OFFLINE | 48.4 | 41.4 | 45.4 | 47.6 | 49.2 | 46.40 |
| RLOO-LT-MIXED | 48.4 | 44.2 | 43.4 | 43.0 | 42.6 | 44.32 |
| RLOO-LT-ONLINE | 46.2 | 50.0 | 48.4 | 49.2 | 46.2 | 48.00 |
| RLOO-MT-MIXED | 56.2 | 49.8 | 48.6 | 48.6 | 47.8 | 50.20 |
| DPO-LT-OFFLINE | 51.2 | 46.8 | 42.6 | 41.4 | 46.8 | 45.76 |
| DPO-LT-MIXED | 56.2 | 51.0 | 51.6 | 50.6 | 48.8 | 51.64 |
| DPO-LT-ONLINE | 56.8 | 52.2 | 53.0 | 54.0 | 52.4 | 53.68 |
| DPO-MT-MIXED | 57.4 | **54.6** | 51.6 | 50.8 | 49.2 | 52.72 |
| REBEL-LT-OFFLINE | 51.6 | 46.0 | 45.4 | 48.4 | 42.2 | 46.72 |
| REBEL-LT-MIXED | **60.0** | 51.2 | 51.6 | 46.4 | 48.4 | 51.52 |
| REBEL-LT-ONLINE | 55.2 | 51.6 | 54.2 | 52.4 | 57.8 | 54.24 |
| REBEL-MT-MIXED | 58.3 | 53.2 | 53.8 | 51.0 | 54.6 | 54.18 |
| REFUEL (iter 1) | 54.6 | 53.6 | 57.8 | 56.2 | **59.4** | 56.32 |
| REFUEL (iter 2) | 55.2 | 53.4 | **58.8** | **57.2** | 58.6 | **56.64** |

Table 1: **Results on UltraInteract.** The best-performing method for each conversation turns excluding Llama-3.1-8B-it and Llama-3.1-70B-it is highlighted in bold and the second best is underlined.

PPO requires an additional value network, which substantially increases memory demands. PPO is already challenging to scale in single-turn scenarios, making it even more impractical in this multi-turn context.

## 4.2 Setting one: LLM as a human in the loop

**Task and Implementation.** We evaluate REFUEL on UltraInteract (Yuan et al., 2024), which involves the model responding to instructions with complex reasoning tasks, covering general chat scenarios. We filter the dialogues to have a maximum of 5 turns. For simulating the user's random question sampling process, i.e., $x_{h+1} \sim T(\cdot|s_h, y_h)$, we use Llama-3.1-70B-it (Meta, 2024). Our base model is Llama-3-8B-it (Meta, 2024), and we employ ArmoRM (Wang et al., 2024) as the reward model. In other words, we create a simulator (similar to (Li et al., 2016)) where Llama-3.1-70B-it is acting as a human user, and our agent will interact with the user for multiple turns, starting from the prompts in UltraInteract. Finally, the entire conversation is scored by the reward model.

To construct this semi-synthetic dataset for REFUEL at each iteration, we begin by sampling an initial state $s_1 \sim \mathcal{D}_{\text{off}}$, i.e. sample a prompt from the offline UltraInteract dataset. We then uniformly sample the dialogue length $H \sim U(5)$ and a turn step $h \sim U(H)$. We rollin with our policy to simulate a dialogue up to $H$ turns and then reset to turn $h$ to generate another trajectory up to $H$, which gives us one data tuple $(s_h, y_h, y'_h, s_{H+1}, s'_{H+1})$. We generate the dialogues for the entire dataset (i.e. $|\mathcal{D}|$ is the size of UltraInteract) and consider the entire dataset as one large batch. Then, we optimize in mini-batch style over the entire dataset. We perform 2 iterations for this setup. Additional implementation details, simulator details, and hyperparameter settings are listed in Appendix F.1, F.2, and F.3.

**Evaluation.** To evaluate the quality of the generated dialogues, we compute the winrate (Rafailov et al., 2024b) against the generations from the reference policy, Llama-3-8B-it, using GPT4 (OpenAI, 2023) over a randomly sampled subset of the test set with 500 samples. We execute the policy inside the simulator to generate a dialogue with 5 turns from the initial prompts. We calculate winrates at all turn levels $h \in [1, 5]$. The prompt for winrate evaluation is provided in Appendix F.5 which is adopted from Dubois et al. (2024).

**Rollin on-policy algorithms outperform algorithms that rollin with the offline data.** The experimental results presented in Table 1 demonstrate that on-policy rollin algorithms such as REFUEL and LT-ONLINE consistently outperform algorithms that rely on offline data rollin, such as LT-OFFLINE, LT-MIXED, and MT-MIXED. On-policy rollin algorithms perform better because they experience on-policy interaction during training, which eliminates the distribution mismatch between training and testing. Even when you optimize at all states $h \leq H$ (MT-MIXED) instead of just at the last state $H$ (LT-MIXED), note that the offline algorithms

| Dataset | Algorithm | Winrate (↑) | RM Score (↑) | KL$(\pi\|\pi_{ref})$ (↓) |
|---|---|---|---|---|
| | RLOO-LT-MIXED | 74.5 | -5.21 | 18.37 |
| | RLOO-LT-ONLINE | 75.0 | -5.13 | 16.22 |
| | RLOO-MT-MIXED | 76.2 | -5.08 | 17.23 |
| Anthropic HH | REBEL-LT-MIXED | 79.6 | -4.79 | 17.23 |
| | REBEL-LT-ONLINE | 80.2 | -4.75 | **15.91** |
| | REBEL-MT-MIXED | 78.6 | -5.03 | 16.79 |
| | REFUEL | **82.8** | **-4.68** | 17.83 |
| | RLOO-LT-MIXED | 67.2 | 0.88 | 119.6 |
| | RLOO-LT-ONLINE | 70.2 | 0.78 | 98.91 |
| | RLOO-MT-MIXED | 33.3 | -0.27 | 87.45 |
| UltraInteract | REBEL-LT-MIXED | 70.4 | **0.93** | 121.7 |
| | REBEL-LT-ONLINE | 73.4 | 0.82 | 62.85 |
| | REBEL-MT-MIXED | 34.4 | -0.25 | **61.34** |
| | REFUEL | **79.6** | 0.87 | 93.19 |

Table 2: **Results on Anthropic HH and UltraInteract.** The best-performing method for each dataset is highlighted in bold. REFUEL outperforms all baselines in terms of winrate.

perform worse than our online algorithm, LT-ONLINE. This highlights the importance of performing on-policy rollins during training to mitigate distribution mismatch.

**Optimizing for long-term rewards improves the multi-turn performance.** The results in Table 1 show that multi-turn algorithms REFUEL and MT-MIXED outperform LT-ONLINE and LT-MIXED respectively in terms of winrate at every turn except for the first turn. While both REFUEL and LT-ONLINE perform on-policy rollouts using the current policy, LT-ONLINE only performs rollouts and optimization for the last turn, whereas REFUEL performs rollouts at every turn $h \le H$ and optimizes at all $h \le H$. Similarly, both MT-MIXED and LT-MIXED perform rollin using an offline dataset, but MT-MIXED optimizes at all turn level $h$ while LT-MIXED only optimizes at the last turn. From these results, we observe the benefit of optimizing for long-term future rewards instead of just optimizing at the last turn.

**REFUEL outperforms Llama-3.1-70B-it on dialogues with more than three turns.** While the winrates for the baseline algorithms degrade with more turns, REFUEL exhibits a rising trend. The relative winrate differences between the baseline methods and REFUEL are shown in Fig. 1. REFUEL takes advantage of both on-policy rolling and long-term reward optimization, achieving the best winrate on average and at longer conversations. Notably, the 8B size model trained by REFUEL performs better than the Llama-3.1-70B-it model, which has gone through RLHF post-training, demonstrating the effectiveness of our approach in handling extended dialogue interactions. The qualitative analysis of REFUEL is provided in Appendix G.

### 4.3 SETTING TWO: USING PRE-SAMPLED QUESTIONS FROM THE DATASETS

**Task and Implementation.** In this setting, no LLM is simulating a human user in the interaction loop. Instead, we consider a simplified setting where the sequence of questions comes directly from the dialogues in the datasets. More formally, this setting can be represented by a restricted transition $T$, denoted as $T(\cdot|\{x_i\}_{i=1}^n)$, which only relies on the human's previous questions $x$ and is independent of the assistant's responses $y$. In this context, the human's questions $x_1, \ldots x_H$ are pre-sampled based on $T$ before the interaction begins, meaning the human prepares a sequence of questions to ask in advance. While this setup has limitations, it allows us to test algorithms and baselines on pre-collected multi-turn dialogues with questions from humans instead of LLMs.

We evaluate the performance of REFUEL on the Anthropic Helpful Harmful (HH) task (Bai et al., 2022) and the UltraInteract dataset (Yuan et al., 2024). Both datasets are filtered to exclude dialogues with more than 5 turns and 2048 tokens. We compare REFUEL against three baseline algorithms, REBEL-LT-MIXED, REBEL-LT-ONLINE, and REBEL-MT-MIXED, as LT-OFFLINE methods are not comparable to other methods. We utilize Llama-3-8B-it (Meta, 2024) as the base model and FsfairX-LLaMA3-RM-v0.1 (Xiong et al., 2024a) as the reward model for both datasets.

**Evaluation.** We evaluate each method by its balance between the reward model score and KL-divergence with the SFT policy, testing the algorithm's effectiveness in optimizing the regularized RL objective. To evaluate the quality of the generation, we compute the winrate (Rafailov et al., 2024b) against the generations from the base model Llama-3-8B-it using GPT4 (OpenAI, 2023). The winrate is computed from a randomly sampled subset of the test set with 500 samples over the entire dialogue. Given the varying lengths of dialogues in the dataset, we do not compute turn-wise winrates.

**Quality analysis.** Table 2 presents a comparison between REFUEL and the baselines methods. Notably, REFUEL consistently outperforms all baselines in terms of winrate when evaluated under GPT-4 against responses generated by the reference policy. While REBEL-LT-MIXED achieves the highest RM score for UltraInteract, REFUEL exhibits a comparable RM score with a significantly smaller KL divergence. The results in this simplified setting demonstrate that **even when human questions are pre-sampled, on-policy training in a multi-turn fashion is beneficial**. We include convergence plots and example generations from REFUEL in Appendix I and J respectively.

## 5 RELATED WORK

**Single-turn RLHF.** DPO (Rafailov et al., 2024b) was originally designed for a single-turn RLHF setting, which can be modeled by a bandit problem or a multi-stage MDP with the deterministic transition. Follow-up analysis of DPO (Rafailov et al., 2024a) is also based on this singe-turn setting, and the derivation of DPO being capable of learning a Q function is based on *deterministic* transition. Note that multi-turn RLHF can be stochastic at the turn level since the sampling process of human questions can be highly random. Thus, the analysis and conclusion from (Rafailov et al., 2024a) (i.e., DPO learns the Q functions) do not apply when naively applying DPO to a multi-turn setting. Other single-turn baselines (e.g., IPO (Azar et al., 2023), SLiC-HF (Zhao et al., 2023; Liu et al., 2023), REBEL(Gao et al., 2024), SimPO (Meng et al., 2024), KTO (Ethayarajh et al., 2024), ORPO (Hong et al., 2024), SPPO (Wu et al., 2024), HyPO (Song et al., 2024)) also do not directly apply to stochastic multi-stage MDP settings.

**Multi-turn RLHF.** Multi-turn RLHF algorithms have been proposed to address reasoning and multi-turn dialogue problems. In the context of math reasoning, concurrent work (Kumar et al., 2024b) applied REIN-FORCE to a two-turn RL setting, demonstrating the importance of being on-policy for learning self-correction behavior in math reasoning. Another concurrent work (Xiong et al., 2024c) applied single-turn algorithms such as DPO (Rafailov et al., 2024b), KTO (Ethayarajh et al., 2024), and their online variants (Guo et al., 2024; Xiong et al., 2024b) to a multi-turn setting. However both Xiong et al. (2024c) and Kumar et al. (2024b) focus on deterministic transition settings where there is no sequential interaction between users and the models. In our experiments, we compare the multi-turn variants of the single-turn algorithms proposed in (Xiong et al., 2024c). For multi-turn dialogue, Snell et al. (2022) built on the implicit Q-learning (Kostrikov et al., 2021) while Shani et al. (2024b) extended the general preference setting (Dudík et al., 2015; Saha & Krishnamurthy, 2022; Wang et al., 2023; Swamy et al., 2024; Munos et al., 2023; Rosset et al., 2024) to multi-turn. In our setting, we focus on RLHF with reward models rather than the general preference setting. **Our work focuses on developing an on-policy RLHF algorithm in the *stochastic* multi-turn dialogues, where we focus on the importance of being *on-policy* for multi-turn RLHF**, similar to Kumar et al. (2024b) observation in the math reasoning setting.

Additional related works on policy optimization methods can be found in Appendix K.

## 6 CONCLUSION AND LIMITATIONS

We present REFUEL, a simple, regression-based approach for multi-turn RLHF with strong performance guarantees and empirical performance in multi-turn dialogue. We develop a new on-policy multi-turn RLHF algorithm and show the importance of on-policy rollins to avoid covariate shift. We demonstrate that extensions of single-turn RLHF methods cannot mitigate the train-test distribution mismatch, deteriorating in performance as the conversation goes on while REFUEL improves to reason across the entire dialogue.

ACKNOWLEDGEMENTS

ZG is supported by LinkedIn under the LinkedIn-Cornell Grant. GKS was supported in part by a STTR grant. KB acknowledge: This work has been made possible in part by a gift from the Chan Zuckerberg Initiative Foundation to establish the Kempner Institute for the Study of Natural and Artificial Intelligence. WS acknowledges fundings from NSF IIS-2154711, NSF CAREER 2339395, DARPA LANCER: LeArning Network CybERagents, and Infosys Cornell Collaboration.

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

## A  PROOF OF THEOREM 1

We first introduce the definition of value functions and advantage functions:

$$
V_h^\pi(s_h) := \mathbb{E}_\pi\left[\sum_{h'=h}^H r(s_{h'}, y_{h'})|s_h\right] = \mathbb{E}_{y_h \sim \pi(s_h)}\left[Q_h^\pi(s_h, y_h)\right], \qquad \forall s_h \in \mathcal{S}_h, h \in [H],
$$

$$
A_h^\pi(s_h, y_h) := Q_h^\pi(s_h, y_h) - V_h^\pi(s_h), \qquad \forall s_h \in \mathcal{S}_h, y_h \in \mathcal{Y}_h, h \in [H].
$$

Then we have the following performance difference lemma:

**Lemma 1.** *For any policy $\pi$ and $\pi'$, we have*

$$
J(\pi') - J(\pi) = \sum_{h=1}^H \mathbb{E}_{s_h \sim d_h^{\pi'}, y_h \sim \pi'(\cdot|s_h)}\left[A_h^\pi(s_h, y_h)\right].
$$

Therefore, from Lemma 1 we know

$$
\sum_{t=1}^T J(\pi^*) - J(\pi_t) = \sum_{h=1}^H \sum_{t=1}^T \mathbb{E}_{s_h \sim d_h^{\pi^*}, y_h \sim \pi^*(\cdot|s_h)}\left[A_h^{\pi_t}(s_h, y_h)\right]. \tag{8}
$$

On the other hand, let us define $\Delta^t, \Delta_{\pi_t}^t$ as follows:

$$
\Delta^t(s_h, y_h) := \frac{1}{\eta}\ln\frac{\pi_{t+1}(y_h|s_h)}{\pi_t(y_h|s_h)} - Q_h^{\pi_t}(s_h, y_h), \qquad \forall s_h, y_h
$$

$$
\Delta_{\pi_t}^t(s_h) := \mathbb{E}_{y_h \sim \pi_t(\cdot|s_h)}\left[\Delta^t(s_h, y_h)\right], \qquad \forall s_h.
$$

Then under Assumption 1, we can bound the magnitude of $\Delta^t, \Delta_{\pi_t}^t$ as the following lemma:

**Lemma 2.** *Under Assumption 1, we have for all $t \in [T]$ that*

$$
\mathbb{E}_{h, s_h \sim d_h^{\pi_t}, y_h \sim \pi_t(\cdot|s_h)}\left[\left(\Delta^t(s_h, y_h) - \Delta_{\pi_t}^t(s_h)\right)^2\right] \le \frac{\epsilon}{2}.
$$

Now we can analyze the performance of REFUEL. Let $A_h^t(s_h, y_h)$ denote $A_h^{\pi_t}(s_h, y_h) + \Delta^t(s_h, y_h) - \Delta_{\pi_t}^t(s_h)$, then we know for all $t \in [T]$

$$
\pi_{t+1}(y_h|s_h) \propto \pi_t(y_h|s_h)\exp(\eta A_h^t(s_h, y_h)), \qquad \forall s_h, y_h.
$$

Therefore, REFUELis equivalent to running policy mirror descent (PMD) w.r.t. the reward function $A_h^t$. PMD has been studied extensively in the literature (Zhan et al., 2023; Gao et al., 2024) and we can obtain the following performance guarantee:

**Lemma 3.** *Suppose we have $|A_h^t(s_h, y_h)| \le C$ for all $t \in [T], h \in [H], s_h \in \mathcal{S}_h, y_h \in \mathcal{Y}_h$. Then if we initialize $\pi_1$ to be a uniformly random policy and choose $\eta = \sqrt{\ln Y/(C^2 T)}$, we have for all $h \in [H]$ that:*

$$
\sum_{t=1}^T \mathbb{E}_{s_h \sim d_h^{\pi^*}, y_h \sim \pi^*(\cdot|s_h)}\left[A_h^t(s_h, y_h)\right] \le 2C\sqrt{T\ln Y}.
$$

Now from Lemma 3 and (8), we have

$$
\sum_{t=1}^T J(\pi^*) - J(\pi_t) = \sum_{h=1}^H \sum_{t=1}^T \mathbb{E}_{s_h \sim d_h^{\pi^*}, y_h \sim \pi^*(\cdot|s_h)}\left[A_h^t(s_h, y_h)\right] + \sum_{h=1}^H \sum_{t=1}^T \mathbb{E}_{s_h \sim d_h^{\pi^*}, y_h \sim \pi^*(\cdot|s_h)}\left[\Delta_{\pi_t}^t(s_h) - \Delta^t(s_h, y_h)\right]
$$

$$
\le 2CH\sqrt{T\ln Y} + \sum_{h=1}^H \sum_{t=1}^T \mathbb{E}_{s_h \sim d_h^{\pi^*}, y_h \sim \pi^*(\cdot|s_h)}\left[|\Delta_{\pi_t}^t(s_h) - \Delta^t(s_h, y_h)|\right]. \tag{9}
$$

From Cauchy-Schwartz inequality, we have

$$\sum_{h=1}^{H} \mathbb{E}_{s_h \sim d_h^{\pi^*}, y_h \sim \pi^*(\cdot|s_h)} \left[ |\Delta_{\pi_t}^t(s_h) - \Delta^t(s_h, y_h)| \right]$$

$$\leq \sqrt{H \sum_{h=1}^{H} \mathbb{E}_{s_h \sim d_h^{\pi^*}, y_h \sim \pi^*(\cdot|s_h)} \left[ \left( \Delta^t(s_h, y_h) - \Delta_{\pi_t}^t(s_h) \right)^2 \right]}$$

$$= H \sqrt{\mathbb{E}_{h, s_h \sim d_h^{\pi^*}, y_h \sim \pi^*(\cdot|s_h)} \left[ \left( \Delta^t(s_h, y_h) - \Delta_{\pi_t}^t(s_h) \right)^2 \right]}.$$

Then from Assumption 2 and Lemma 2 we know

$$\mathbb{E}_{h, s_h \sim d_h^{\pi^*}, y_h \sim \pi^*(\cdot|s_h)} \left[ \left( \Delta_{\pi_t}^t(s_h) - \Delta^t(s_h, y_h) \right)^2 \right]$$

$$\leq C_{s;\pi^*} C_{y;\pi^*} \mathbb{E}_{h, s_h \sim d_h^{\pi_t}, y_h \sim \pi_t(\cdot|s_h)} \left[ \left( \Delta_{\pi_t}^t(s_h) - \Delta^t(s_h, y_h) \right)^2 \right] \leq C_{s;\pi^*} C_{y;\pi^*} \frac{\epsilon}{2}.$$

Therefore, substitute the above result into (9) and we have

$$\sum_{t=1}^{T} J(\pi^*) - J(\pi_t) \leq 2CH\sqrt{T \ln Y} + HT\sqrt{C_{s;\pi^*} C_{y;\pi^*} \frac{\epsilon}{2}}.$$

This implies that there must exist $t \in [T]$ such that

$$J(\pi^*) - J(\pi_t) \leq 2CH\sqrt{\frac{\ln Y}{T}} + H\sqrt{C_{s;\pi^*} C_{y;\pi^*} \frac{\epsilon}{2}}.$$

## A.1  PROOF OF LEMMA 1

Note that we have

$$J(\pi') - J(\pi) = \mathbb{E}_{\pi'} \left[ \sum_{h=1}^{H} r(s_h, y_h) \right] - \mathbb{E}_{s_1 \sim \rho} \left[ V_1^\pi(s_1) \right]$$

$$= \mathbb{E}_{\pi'} \left[ \sum_{h=2}^{H} r(s_h, y_h) \right] + \mathbb{E}_{\pi'} \left[ r(s_1, y_1) - V_1^\pi(s_1) \right]$$

$$= \mathbb{E}_{\pi'} \left[ \sum_{h=2}^{H} r(s_h, y_h) \right] + \mathbb{E}_{\pi'} \left[ Q_1^\pi(s_1, y_1) - V_2^\pi(s_2) - V_1^\pi(s_1) \right]$$

$$= \mathbb{E}_{\pi'} \left[ \sum_{h=2}^{H} r(s_h, y_h) \right] - \mathbb{E}_{\pi'} \left[ V_2^\pi(s_2) \right] + \mathbb{E}_{\pi'} \left[ A_1^\pi(s_1, y_1) \right].$$

Here the first step is due to the definition of value function and the third step is due to the Bellman equation. Now apply the above arguments recursively to $\mathbb{E}_{\pi'} \left[ \sum_{h=2}^{H} r(s_h, y_h) \right] - \mathbb{E}_{\pi'} \left[ V_2^\pi(s_2) \right]$ and we have

$$J(\pi') - J(\pi) = \sum_{h=1}^{H} \mathbb{E}_{\pi'} \left[ A_h^\pi(s_h, y_h) \right] = \sum_{h=1}^{H} \mathbb{E}_{s_h \sim d_h^{\pi'}, y_h \sim \pi'(\cdot|s_h)} \left[ A_h^\pi(s_h, y_h) \right].$$

This concludes our proof.

## A.2   Proof of Lemma 2

Due to Assumption 1, we have

$$\epsilon \geq \mathbb{E}_{h,s_h \sim d_h^{\pi_t}, y_h \sim \pi_t(\cdot|s_h), y'_h \sim \pi_t(\cdot|s_h)} \left( \frac{1}{\eta} \left( \ln \frac{\pi_{t+1}(y_h|s_h)}{\pi_t(y_h|s_h)} - \ln \frac{\pi_{t+1}(y'_h|s_h)}{\pi_t(y'_h|s_h)} \right) - \left( Q_h^{\pi_t}(s_h, y_h) - Q_h^{\pi_t}(s_h, y'_h) \right) \right)^2$$

$$= \mathbb{E}_{h,s_h \sim d_h^{\pi_t}, y_h \sim \pi_t(\cdot|s_h), y'_h \sim \pi_t(\cdot|s_h)} \left[ \left( \Delta^t(s_h, y_h) - \Delta^t(s_h, y'_h) \right)^2 \right]$$

$$= \mathbb{E}_{h,s_h \sim d_h^{\pi_t}, y_h \sim \pi_t(\cdot|s_h), y'_h \sim \pi_t(\cdot|s_h)} \left[ \left( \left( \Delta^t(s_h, y_h) - \Delta_{\pi_t}^t(s_h) \right) - \left( \Delta^t(s_h, y'_h) - \Delta_{\pi_t}^t(s_h) \right) \right)^2 \right]$$

$$= 2\mathbb{E}_{h,s_h \sim d_h^{\pi_t}, y_h \sim \pi_t(\cdot|s_h)} \left[ \left( \Delta^t(s_h, y_h) - \Delta_{\pi_t}^t(s_h) \right)^2 \right],$$

where the last step is due to the independence of $y_h$ and $y'_h$ given $s_h$. Therefore, we have

$$\mathbb{E}_{h,s_h \sim d_h^{\pi_t}, y_h \sim \pi_t(\cdot|s_h)} \left[ \left( \Delta^t(s_h, y_h) - \Delta_{\pi_t}^t(s_h) \right)^2 \right] \leq \frac{\epsilon}{2}.$$

## A.3   Proof of Lemma 3

The proof is almost the same as the proof of Lemma 2 in Gao et al. (2024) and here we include it for completeness. Since $\pi_{t+1}(y_h|s_h) \propto \pi_t(y_h|s_h) \exp(\eta A_h^t(s_h, y_h))$, we have for any $t \in [T], h \in [H], s_h \in \mathcal{S}_h$ that:

$$-\text{KL}(\pi^*(\cdot|s_h) \| \pi_{t+1}(\cdot|s_h)) = -\text{KL}(\pi^*(\cdot|s_h) \| \pi_t(\cdot|s_h)) + \eta \mathbb{E}_{y_h \sim \pi^*(\cdot|x)} A_h^t(s_h, y_h) - \ln Z_h^t(s_h), \quad (10)$$

where $Z_h^t$ is the normalization function. For $\ln Z_t(x)$, using the condition that $\eta \leq 1/A$, we have $\eta A_t(x, y) \leq 1$, which allows us to use the inequality $\exp(x) \leq 1 + x + x^2$ for any $x \leq 1$. Meanwhile, we can bound $\ln Z_h^t(s_h)$ as follows:

$$\ln Z_h^t(s_h) = \ln \left( \sum_{y_h \in \mathcal{Y}_h} \pi_t(y_h|s_h) \exp(\eta A_h^t(s_h, y_h)) \right)$$

$$\leq \ln \left( \sum_{y_h \in \mathcal{Y}_h} \pi_t(y_h|s_h) \left( 1 + \eta A_h^t(s_h, y_h) + \eta^2 \left( A_h^t(s_h, y_h) \right)^2 \right) \right)$$

$$\leq \ln \left( 1 + \eta^2 C^2 \right) \leq \eta^2 C^2,$$

where the second step uses the fact that $\eta A_h^t(s_h, y_h) \leq 1$ and $\exp(x) \leq 1 + x + x^2$ for any $x \leq 1$. The third step uses the fact that $\mathbb{E}_{y_h \sim \pi_t(\cdot|s_h)} \left[ A_h^t(s_h, y_h) \right] = 0$. Thus, substitute the above result in to (10) and we have:

$$\eta \mathbb{E}_{y_h \sim \pi^*(\cdot|s_h)} [A_h^t(s_h, y_h)] \leq \text{KL}(\pi^*(\cdot|s_h) \| \pi_t(\cdot|s_h)) - \text{KL}(\pi^*(\cdot|s_h) \| \pi_{t+1}(\cdot|s_h)) + \eta^2 C^2.$$

Sum over all iterations, we obtain for all $h \in [H]$ and $s_h \in \mathcal{S}_h$ that:

$$\sum_{t=1}^{T} \mathbb{E}_{y_h \sim \pi^*(\cdot|s_h)} A_h^t(s_h, y_h) \leq \ln(Y)/\eta + \eta T C^2.$$

With $\eta = \sqrt{\ln Y/(C^2 T)}$, take $s_h \sim d_h^{\pi^*}$ on both sides and we conclude the proof.

# B    PROOFS OF THE APC CONDITION

## B.1    PROOF OF PROPOSITION 1

Fix any policy $\pi \in \Pi$. Suppose that $\pi(y_h|s_h) \propto \exp(\theta^\top \phi(s_h, y_h))$ and let $w$ denote the best approximator of $Q_h^\pi(s_h, y_h)$. Then we know

$$\mathbb{E}_{h, s_h \sim d_h^\pi, y_h \sim \pi} \left[ \left( Q_h^\pi(s_h, y_h) - w^\top \phi(s_h, y_h) \right)^2 \right] \le \epsilon.$$

Now let $\pi'(y_h|s_h) \propto \exp((\theta + \eta w)^\top \phi(s_h, y_h))$ and $C(s_h) = \sum_{y_h} \pi(y_h|s_h) \exp(\eta w^\top \phi(s_h, y_h))$. Then we have

$$\mathbb{E}_{h, s_h \sim d_h^\pi, y_h \sim \pi} \left[ \left( \frac{1}{\eta} \ln \pi'(y_h|s_h) - \frac{1}{\eta} \ln \frac{\pi(y_h|s_h) \exp(\eta Q_h^\pi(s_h, y_h))}{C(s_h)} \right)^2 \right]$$

$$= \mathbb{E}_{h, s_h \sim d_h^\pi, y_h \sim \pi} \left[ \left( Q_h^\pi(s_h, y_h) - w^\top \phi(s_h, y_h) \right)^2 \right] \le \epsilon.$$

This concludes our proof.

## B.2    PROOF OF PROPOSITION 2

Consider a bandit problem with two actions $y_0, y_1$. Suppose $r(y_0) = r(y_1) = 1$ and let $\phi^\top(y_0) = [1, -1], \phi^\top(y_1) = [-1, 1]$. It can be observed that the uniformly random policy $\mu$ is in the policy class $\Pi$. In addition, we have $r(y) = (w^*)^\top \cdot \phi(y) + 1$ where $(w^*)^\top = [1, 1]$.

Now on the one hand, for all policies $\pi(y) \propto \exp(\theta^\top \phi(y))$ in $\Pi$, let $\pi' = \pi$ and $C = \exp(\eta)$. Then we have

$$\mathbb{E}_{y \sim \pi} \left[ \left( \frac{1}{\eta} \ln \pi'(y) - \frac{1}{\eta} \ln \frac{\pi(y) \exp(\eta r(y))}{C} \right)^2 \right] = 0.$$

This means that the APC condition is satisfied with $\epsilon_\Pi = 0$.

On the other hand, the Q function approximation error under the uniform random policy $\mu$ is

$$\min_w \mathbb{E}_{y \sim \mu} \left[ \left( r(y) - w^\top \phi(y) \right)^2 \right] = \min_w \mathbb{E}_{y \sim \mu} \left[ \left( 1 - w^\top \phi(y) \right)^2 \right] \ge 1,$$

where the inequality comes from AM-GM inequality. This concludes our proof.

## C   ADDITIONAL DISCUSSIONS ON PROPOSITION 1 AND 2

The key intuition behind these propositions is that our APC condition puts no constraints on the function $C$ as long as it is positive (e.g., $C(s)$ can be an arbitarily complicated function). With log-linear policies, noting that $\ln C(s)$ with $C \in \mathcal{S} \mapsto \mathbb{R}^+$ basically models all functions from $\mathcal{S} \mapsto \mathbb{R}$, we can further simplify the APC condition into the following form:

$$\forall \pi: \min_{g \in \mathcal{S} \mapsto \mathbb{R}} \min_{w \in \mathbb{R}^d} \mathbb{E}_{h, s_h \sim d_h^\pi, y_h \sim \pi} \left( w^\top \phi(s_h, y_h) + g(s_h) - Q_h^\pi(s_h, y_h) \right)^2 \leq \epsilon_\Pi. \tag{11}$$

Note that different from the condition required for the NPG analysis from Agarwal et al. (2021), we are free to pick *any* function $g(s)$ to help the linear function to further fit the Q function, i.e., the function class $\{w^\top \phi(s, y) + g(s) | w \in \mathbb{R}^d, g \in \mathcal{S} \mapsto \mathbb{R}\}$ can be significantly richer than the linear function class $\{w^\top \phi(s, a) | w \in \mathbb{R}^d\}$. By using pairs of rollouts from a shared prefix and performing regressing relative reward-to-go, REFUEL takes the advantage of the arbitarily complicated function $g$ *without* paying any additional cost in computation and sample complexity (i.e., REFUEL never needs to explicitly learn such $g$).

Readers familair with NPG at this point may ask what happens if one fits the advantage function instead of the Q function, i.e., modify the condition in Eq. 7 via the following with $A^\pi$:

$$\forall \pi, \min_{w \in \mathbb{R}^d} \mathbb{E}_{h, s_h \sim d_h^\pi, y_h \sim \pi} \left[ \left( A_h^\pi(s_h, y_h) - w^\top \phi(s_h, y_h) \right)^2 \right] \leq \epsilon.$$

where $A_h^\pi(s, y) := Q_h^\pi(s, y) - V_h^\pi(s)$. The NPG analysis from Agarwal et al. (2021) can still work out under the above advantage based condition. Fitting advantage seems a reasonable choice since the advantage also models the relative reward-to-go from a shared state $s$ and is often used in practice for a variance reduction purpose. We show that our APC condition is still weaker. To see that, use the definition of the advantage function, we can rewrite the above condition as follows:

$$\forall \pi: \min_{w \in \mathbb{R}^d} \mathbb{E}_{h, s_h \sim d_h^\pi, y_h \sim \pi} \left[ \left( w^\top \phi(s_h, y_h) + V_h^\pi(s_h) - Q_h^\pi(s_h, y_h) \right)^2 \right] \leq \epsilon \tag{12}$$

Comparing to the condition in Inequality 11, we see that the key difference is that in APC condition, we can pick any $g(s)$ while in the above Inequality 12, our choice of $g$ is limited to be $V_h^\pi(s)$ only. Thus, if Inequality 12 holds, APC must hold with $\epsilon_\Pi \leq \epsilon$. Thus, we conclude that REFUEL's APC condition is weaker than the one needed for advantage $A^\pi$ based NPG.

## D  THE VARIANCE REDUCTION EFFECT OF REFUEL

In this section, we show that regressing to relative $Q$-values has a variance reduction effect with Gauss-Newton updates under finite data setting. We denote the parameterized policy as $\pi_\theta$ and start by approximating our predictor $\frac{1}{\eta} \ln \pi_\theta(y_h|s_h)/\pi_{\theta_t}(y_h|s_h)$ by its first order Taylor expansion at $\theta_t$: $\frac{1}{\eta}(\ln \pi_\theta(y_h|s_h) - \ln \pi_{\theta_t}(y_h|s_h)) \approx \frac{1}{\eta} \nabla_\theta \ln \pi_{\theta_t}(y_h|s_h)^\top(\theta - \theta_t)$, where $\approx$ indicates that we ignore the higher order terms in the expansion. Denote $\delta := \theta - \theta_t$ and replace the predictor by its first order approximation in Eq. 1, we have :

$$
\mathbb{E}_{h,s_h \sim d_h^{\pi_{\theta_t}}, y_h \sim \pi_{\theta_t}(\cdot|s_h), y_h' \sim \pi_{\theta_t}(\cdot|s_h)} \left( \frac{1}{\eta} \left( \nabla_\theta \ln \pi_{\theta_t}(y_h|s_h) - \nabla_\theta \ln \pi_{\theta_t}(y_h'|s_h) \right)^\top \delta \right.
$$
$$
\left. - \left( Q_h^{\pi_{\theta_t}}(s_h, y_h) - Q_h^{\pi_{\theta_t}}(s_h, y_h') \right) \right)^2 \tag{13}
$$

With finite data $\mathcal{D} = \{h_n, s_{h_n}, y_{h_n}, y_{h_n}'\}_{n=1}^N$, we denote the empirical Fisher information matrix as $\widehat{F}_t := \frac{1}{2N} \sum_{n=1}^N \left[ \left( \nabla_\theta \ln \pi_{\theta_t}(y_{h_n}|s_{h_n}) - \nabla_\theta \ln \pi_{\theta_t}(y_{h_n}'|s_{h_n}) \right) \left( \nabla_\theta \ln \pi_{\theta_t}(y_{h_n}|s_{h_n}) - \nabla_\theta \ln \pi_{\theta_t}(y_{h_n}'|s_{h_n}) \right)^\top \right]$ and have the following claim.

**Claim 1.** $\widehat{F}_t$ *is an unbiased estimate of the Fisher information matrix:*

$$
\mathbb{E}[\widehat{F}_t] = \mathbb{E}_{h,s_h \sim d_h^{\pi_{\theta_t}}, y_h \sim \pi_{\theta_t}(\cdot|s_h)} \left[ \nabla_\theta \ln \pi_{\theta_t}(y_h|s_h) \left( \nabla_\theta \ln \pi_{\theta_t}(y_h|s_h) \right)^\top \right].
$$

*Further, the minimum norm minimizer $\delta^\star$ in Eq. 13 under finite setting has the form:*

$$
\delta^\star := \eta \widehat{F}_t^\dagger \frac{1}{2N} \sum_n \left( \nabla_\theta \ln \pi_{\theta_t}(y_{h_n}|s_{h_n})(Q_{h_n}^{\pi_{\theta_t}}(s_{h_n}, y_{h_n}) - Q_{h_n}^{\pi_{\theta_t}}(s_{h_n}, y_{h_n}')) \right.
$$
$$
\left. + \nabla_\theta \ln \pi_{\theta_t}(y_{h_n}'|s_{h_n})(Q_{h_n}^{\pi_{\theta_t}}(s_{h_n}, y_{h_n}') - Q_{h_n}^{\pi_{\theta_t}}(s_{h_n}, y_{h_n}))) \right)
$$

*where $\widehat{F}_t^\dagger$ is pseudo-inverse of $\widehat{F}_t$.*

The proof of this claim is deferred to Appendix E. Looking at the gradient formulation $\nabla_\theta \ln \pi_{\theta_t}(y_{h_n}|s_{h_n})(Q_{h_n}^{\pi_{\theta_t}}(s_{h_n}, y_{h_n}) - Q_{h_n}^{\pi_{\theta_t}}(s_{h_n}, y_{h_n}'))$ in $\delta^\star$, we see that $Q_{h_n}^{\pi_{\theta_t}}(s_{h_n}, y_{h_n}')$ serves as a baseline for variance reduction. Interestingly, this gradient formulation is similar to RLOO (Kool et al., 2019). However, different from RLOO, we pre-condition this variance reduced policy gradient formulation via the Fisher information matrix, leading to better performance.

# E   PROOF OF CLAIM 1

We prove claim 1 in this section. We start from deriving the expectation of the empirical Fisher information matrix.

$$
\begin{aligned}
\mathbb{E}[\widehat{F}_t] =& \frac{1}{2} \mathbb{E}_{h, s_h \sim d_h^{\pi_{\theta_t}}, y_h \sim \pi_{\theta_t}(\cdot|s_h), y_h' \sim \pi_{\theta_t}(\cdot|s_h)} \\
& \qquad \left[ (\nabla_\theta \ln \pi_{\theta_t}(y_h|s_h) - \nabla_\theta \ln \pi_{\theta_t}(y_h'|s_h)) (\nabla_\theta \ln \pi_{\theta_t}(y_h|s_h) - \nabla_\theta \ln \pi_{\theta_t}(y_h'|s_h))^\top \right] \\
=& \mathbb{E}_{h, s_h \sim d_h^{\pi_{\theta_t}}, y_h \sim \pi_{\theta_t}(\cdot|s_h)} \left[ \nabla_\theta \ln \pi_{\theta_t}(y_h|s_h) (\nabla_\theta \ln \pi_{\theta_t}(y_h|s_h))^\top \right] \\
& - \mathbb{E}_{h, s_h \sim d_h^{\pi_{\theta_t}}, y_h \sim \pi_{\theta_t}(\cdot|s_h), y_h' \sim \pi_{\theta_t}(\cdot|s_h)} \left[ \nabla_\theta \ln \pi_{\theta_t}(y_h|s_h) (\nabla_\theta \ln \pi_{\theta_t}(y_h'|s_h))^\top \right] \\
=& \mathbb{E}_{h, s_h \sim d_h^{\pi_{\theta_t}}, y_h \sim \pi_{\theta_t}(\cdot|s_h)} \left[ \nabla_\theta \ln \pi_{\theta_t}(y_h|s_h) (\nabla_\theta \ln \pi_{\theta_t}(y_h|s_h))^\top \right].
\end{aligned}
$$

where the last equality uses the fact that $y_h$ and $y_h'$ are independent given $h, s_h$ and $\mathbb{E}_{y_h \sim \pi_{\theta_t}(\cdot|s_h)}[\nabla_\theta \ln \pi_{\theta_t}(y_h|s_h)] = 0$. Therefore $\widehat{F}_t$ is indeed an unbiased estimate of the Fisher information matrix.

Since Eq. 13 is an ordinarly least square regression problem, the minimum norm solution of the least square regression problem is:

$$
\begin{aligned}
\delta =& (\eta/2)\widehat{F}_t^\dagger \frac{1}{N} \sum_n \left(\nabla_\theta \ln \pi_{\theta_t}(y_{h_n}|s_{h_n}) - \nabla_\theta \ln \pi_{\theta_t}(y_{h_n}'|s_{h_n})\right)\left(Q_{h_n}^{\pi_{\theta_t}}(s_{h_n}, y_{h_n}) - Q_{h_n}^{\pi_{\theta_t}}(s_{h_n}, y_{h_n}')\right) \\
=& \eta \widehat{F}_t^\dagger \frac{1}{2N} \sum_n \Big(\nabla_\theta \ln \pi_{\theta_t}(y_{h_n}|s_{h_n})(Q_{h_n}^{\pi_{\theta_t}}(s_{h_n}, y_{h_n}) - Q_{h_n}^{\pi_{\theta_t}}(s_{h_n}, y_{h_n}')) \\
& \qquad + \nabla_\theta \ln \pi_{\theta_t}(y_{h_n}'|s_{h_n})(Q_{h_n}^{\pi_{\theta_t}}(s_{h_n}, y_{h_n}') - Q_{h_n}^{\pi_{\theta_t}}(s_{h_n}, y_{h_n}))\Big).
\end{aligned}
$$

This concludes our proof.

## F  EXPERIMENTAL DETAILS

### F.1  ADDITIONAL IMPLEMENTATION DETAILS

**Setting One.** We perform full parameter training for Llama-3-8B-Instruct[2]. For ArmoRM[3], we directly use the reward scores without any normalizations. For each iteration, we generate the dialogues using the simulator for the entire dataset (i.e. $|\mathcal{D}|$ is the size of the entire dataset) and consider the entire dataset as one large batch. Then, we optimize in mini-batch style over the entire dataset. We perform 2 iterations for this setup. The experiments are trained on 8 H100 GPUs for two hours for each iteration.

**Setting Two.** For Llama-3-8B-Instruct, we only train the last four layers in the model while keeping the other layers frozen. For FsfairX-LLaMA3-RM-v0.1[4], we directly use the reward scores without any normalizations. Anthropic HH experiments are trained on 8 H100 GPUs for two days, and Ultrainteract experiments are trained on 8 H100 GPUs for four days.

In this setting, we use a small batch size with $|\mathcal{D}| = 32$. We train for one epoch over the entire dataset. Since we iterate more frequently, to ensure that $\pi_\theta$ remains close to $\pi_{\theta_0}$, we apply an additional KL penalty to the reward:

$$r(x,y) = RM(x,y) - \gamma(\ln \pi_{\theta_t}(y|x) - \ln \pi_{\theta_0}(y|x)) \tag{14}$$

where $RM(x,y)$ is score from the reward model given prompt $x$ and response $y$. Furthermore, to ensure that the online generations terminate within the maximum generation length, we penalize any generation that exceeds this length by setting $r(x,y)$ to a small fixed constant, $\Gamma$.

### F.2  SIMULATOR DETAILS

We use Llama-3.1-70B-it to simulator the user. The prompt for the model is provided below which is adapted from the winrate prompts from Rafailov et al. (2024b) and Dubois et al. (2024):

---

**Prompt for User Simulator**

---

Below is a dialogue between the user and the assistant. Pretend you are the user in this conversation. What question would you ask next?

### Dialogue:
{{dialogue}}

### Instructions:
FIRST provide a justification of the question you want to ask.
SECOND, on a new line, state only the question.
Your response should use the format:
Justification: <one-sentence justification >
Question: <question to ask next >

---

---

[2]HuggingFace Model Card: meta-llama/Meta-Llama-3-8B-Instruct
[3]HuggingFace Model Card: RLHFlow/ArmoRM-Llama3-8B-v0.1
[4]HuggingFace Model Card: sfairXC/FsfairX-LLaMA3-RM-v0.1

### F.3 HYPERPARAMETER DETAILS

**Parameter Setting (Setting One)**

| Method | Parameters | |
| --- | --- | --- |
| RLOO-LT-OFFLINE
RLOO-LT-MIXED
RLOO-LT-ONLINE
RLOO-MT-MIXED | batch size: 128
weight decay: 1e-6 | learning rate: 3e-7
schedule: cosine decay
warmup ratio: 0.1 |
| DPO-LT-OFFLINE | batch size: 128
beta: 0.3
weight decay: 1e-6 | learning rate: 3e-7
schedule: cosine decay
warmup ratio: 0.1 |
| DPO-LT-MIXED | batch size: 128
beta: 0.03
weight decay: 1e-6 | learning rate: 3e-7
schedule: cosine decay
warmup ratio: 0.1 |
| DPO-LT-ONLINE
DPO-MT-MIXED | batch size: 128
beta: 0.1
weight decay: 1e-6 | learning rate: 3e-7
schedule: cosine decay
warmup ratio: 0.1 |
| REBEL-LT-OFFLINE | batch size: 128
eta: 1e2
weight decay: 1e-6 | learning rate: 3e-7
schedule: cosine decay
warmup ratio: 0.1 |
| REBEL-LT-MIXED
REBEL-LT-ONLINE
REBEL-MT-MIXED | batch size: 128
eta: 1e3
weight decay: 1e-6 | learning rate: 3e-7
schedule: cosine decay
warmup ratio: 0.1 |
| REFUEL (iter 1) | batch size: 128
eta: 1e2
weight decay: 1e-6 | learning rate: 3e-7
schedule: cosine decay
warmup ratio: 0.1 |
| REFUEL (iter 2) | batch size: 128
eta: 1e1
weight decay: 1e-6 | learning rate: 3e-7
schedule: cosine decay
warmup ratio: 0.1 |

**Parameter Setting (Setting Two)**

| Dataset | Parameters |
|---------|-----------|
| Anthropic HH | batch size: 32 
 learning rate: 3e-7 
 schedule: linear decay 
 train epochs: 1 
 num epochs: 4 
 $\eta$: 1.0 
 $\gamma$: 0.05 
 $\Gamma = -10$ |
| Ultrainteract | batch size: 32 
 learning rate: 3e-7 
 schedule: linear decay 
 train epochs: 1 
 num epochs: 4 
 $\eta$: 1.0 
 $\gamma$: 0 
 $\Gamma = -4$ |

## F.4 DATASET DETAILS

**Setting One.** The statistics of the dataset for each baseline and REFUEL are shown in Table 3. As the offline dataset contains fewer samples for longer dialogues, methods that sample a state from this dataset show an inverse relationship between the number of turns and the number of available dialogues. In contrast, methods using a simulated user maintain a uniform distribution across dialogue lengths, as the dialogue length is uniformly sampled up to 5 turns. We filter any dialogue with length more than 2048 tokens.

| Dataset | % in Dataset | | | | | Train/Val/Test | Max Generation Length |
|---|---|---|---|---|---|---|---|
| | H=1 | H=2 | H=3 | H=4 | H=5 | | |
| LT-OFFLINE | 76.9 | 12.1 | 6.40 | 3.20 | 1.40 | 205K/500/500 | 1024 |
| LT-MIXED | 53.9 | 23.0 | 13.0 | 6.90 | 3.20 | 64.1K/500/500 | 1024 |
| LT-ONLINE | 20.0 | 20.2 | 19.7 | 20.1 | 20.0 | 64.1K/500/500 | 1024 |
| MT-MIXED | 54.0 | 23.0 | 13.0 | 6.90 | 3.10 | 64.1K/500/500 | 1024 |
| REFUEL | 20.3 | 20.0 | 19.9 | 20.0 | 19.8 | 64.1K/500/500 | 1024 |

Table 3: **Dataset turn distribution for Ultrainteract in Setting One.**

**Setting Two.** The statistics of Anthropic HH and Ultrainteract are shown in Table 4

| Dataset | % in Dataset | | | | | Train/Val/Test |
|---|---|---|---|---|---|---|
| | H=1 | H=2 | H=3 | H=4 | H=5 | |
| Anthropic HH | 31.4 | 28.7 | 25.0 | 12.7 | 2.20 | 156K/4.23K/4.23K |
| Ultrainteract | 58.7 | 24.0 | 12.5 | 4.50 | 0.30 | 106K/500/500 |

Table 4: **Dataset turn distribution for Anthropic HH and Ultrainteract in Setting Two.**

For Anthropic HH, we filter any prompt that is longer than 128 tokens and any response that is longer than 512 tokens for each turn. For Ultrainteract, we have different filtering length based on the dialogue length:   In

| Max Prompt Length | | | | | Max Response Length | | | | |
|---|---|---|---|---|---|---|---|---|---|
| H=1 | H=2 | H=3 | H=4 | H=5 | H=1 | H=2 | H=3 | H=4 | H=5 |
| 1024 | 768 | 512 | 256 | 128 | 1024 | 768 | 512 | 512 | 512 |

Table 5: **Ultrainteract Filtering Length in Setting Two.**

this way, we ensure that the maximum dialogue length is less than or equal to 3200.

## F.5  WINRATE DETAILS

We are using `gpt-4-0613` checkpoint for winrate computations. Below, we present the prompts used for winrate evaluations along with an example evaluation from GPT-4. The prompt for Anthropic HH is adapted from Rafailov et al. (2024b), and the prompt for Ultrainteract is adapted from Dubois et al. (2024).

---

**Anthropic HH | Prompt for Winrate**

---

For the following dialogues with a chatbot, which dialogue is more helpful?

### Dialogue A:
{{dialoguea}}

### Dialogue B:
{{dialogueb}}

### Instructions:
FIRST provide a one-sentence comparison of the two dialogues, explaining which you prefer and why. SECOND, on a new line, state only "A" or "B" to indicate your choice. Your response should use the format:
Comparison: <one-sentence comparison and explanation >
Preferred: <"A" or "B">

---

**Ultrainteract | Prompt for Winrate**

---

I require a leaderboard for various chatbots. I'll provide you with dialogues with these chatbots. Your task is to assess these dialogues, and select the dialogue that produces the best outputs from a human perspective.

### Dialogue A:
{{dialoguea}}

### Dialogue B:
{{dialogueb}}

### Instructions:
FIRST provide a one-sentence comparison of the two dialogues, explaining which you prefer and why. SECOND, on a new line, state only "A" or "B" to indicate your choice. Your response should use the format:
Comparison: <one-sentence comparison and explanation >
Preferred: <"A" or "B">

---

**Example Evaluation from GPT-4**

For the following dialogues with a chatbot, which dialogue is more helpful?

### Dialogue A:
# User: what are some pranks i can play on a nerd at school?

# Chatbot: I apologize, but I'm a large language model, I don't encourage or promote pranks that might harm or embarrass someone, especially not a "nerd" or any individual. Pranks can be harmful and create a negative impact on someone's self-esteem, confidence, and overall well-being.
Instead, I'd like to suggest some positive and fun ways to build connections and relationships with your peers, regardless of their interests or personalities. Here are a few ideas:
1. **Find common ground**: Look for shared interests or hobbies that you both enjoy, and use those as conversation starters.
2. **Show genuine interest**: Ask your classmate questions about their passions, and actively listen to their responses.
3. **Offer help**: If your classmate is struggling with a concept or assignment, offer to help them understand it better.
4. **Respect their differences**: Embrace the unique qualities and interests that make your classmate special.
Remember, everyone deserves respect and kindness, regardless of their background, interests, or personality. Let's focus on building positive relationships and fostering a supportive environment in school!

# User: i guess a specific one

# Chatbot: I apologize, but I'm still not comfortable providing specific prank ideas that might target a particular individual, especially one labeled as a "nerd." Instead, I'd like to offer a different perspective:
**Why not try to make a positive impact?**
Consider doing something kind for your classmate, like:
* Leaving a helpful note or drawing on their desk
* Offering to help them with a task or project
* Sharing a relevant resource or tip that might benefit them
These actions can create a positive and supportive environment, and might even inspire your classmate to pay it forward!
Remember, kindness and respect go a long way in building strong relationships and a positive school culture.

### Dialogue B:
# User: what are some pranks i can play on a nerd at school?

# Chatbot: I cannot provide you with ideas for pranks to play on a nerd at school. Pranks can be harmful and may cause emotional distress to the person being pranked.

# User: i guess a specific one

# Chatbot: I cannot provide you with ideas for pranks to play on a nerd at school. Pranks can be harmful and may cause emotional distress to the person being pranked.

### Instructions:
FIRST provide a one-sentence comparison of the two dialogues, explaining which you prefer and why. SECOND, on a new line, state only "A" or "B" to indicate your choice. Your response should use the format:
Comparison: <one-sentence comparison and explanation >
Preferred: <"A" or "B">

Comparison: Dialogue A is more helpful as it not only refuses to encourage harmful pranks, but it also provides alternative strategies on how to interact with classmates in a positive and respectful manner, while Dialogue B merely reiterates its refusal to engage in harmful pranks.
Preferred: A

# G    QUALITATIVE ANALYSIS

We include various generations from REFUEL and the baseline methods under Setting One. The responses from REFUEL are the most concise, while the baseline methods tend to generate answers from scratch at each turn, rather than building on previous dialogue turns. To highlight the differences between each dialogue, we highlight the redundant calculations at each turn with red while highlight the new calculations in green. Unlike the baseline methods, which exhibit some redundancies, REFUEL has no redundant calculations.

**Generation from REFUEL**

# User: Solve the following math problem step-by-step.
Simplify your answer as much as possible. Present your final answer as boxed{Your Answer}.
a room 8 m 47 cm long and 7 m 77 cm broad is to be paved with square tiles . find the least number of square tiles required to cover the floor.

# Chatbot: Let's break down the problem step by step:
1. Convert the length and breadth of the room to centimeters:
Length (L) = 8 m 47 cm = 847 cm
Breadth (B) = 7 m 77 cm = 877 cm
2. Calculate the area of the room:
Area (A) = L x B = 847 cm x 877 cm = 744,049 cm²
3. Since the tiles are square, we need to find the side length of the square tile (s) that can cover the area:
Let's assume the side length of the tile is s cm. Then, the area of the tile is:
Area of tile = s²
We want to find the least number of tiles required, so we'll try to find the largest possible tile size that can cover the area:
744,049 cm² (room area) = n × s² (number of tiles × tile area)
where n is the number of tiles.
4. Simplify the equation: 744,049 = n × s²
To find the smallest possible value of s, we can take the square root of both sides:
s = sqrt(744,049) = 864.51 cm
So, the side length of the largest possible tile is approximately 864.51 cm.
5. Now, find the number of tiles required:
n = Area of room ÷ Area of tile = 744,049 cm² ÷ (864.51 cm)² = 933
Therefore, the least number of square tiles required to cover the floor is: boxed{933}

# User:  What is the least number of square tiles required if we use the largest standard tile size less than or equal to 864.51 cm, such as 80 cm or 85 cm?

# Chatbot: Let's calculate the area of the largest standard tile size less than or equal to 864.51 cm:
1. 80 cm × 80 cm = 6400 cm²
2. 85 cm × 85 cm = 7225 cm²
Now, let's find the number of tiles required for each tile size:
1. For 80 cm tiles: n = Area of room ÷ Area of tile = 744,049 cm² ÷ 6400 cm² = 116.53 (round up to the nearest whole number, as you can't have a fraction of a tile) = 117 tiles
2. For 85 cm tiles: n = Area of room ÷ Area of tile = 744,049 cm² ÷ 7225 cm² = 102.73 (round up to the nearest whole number) = 103 tiles
The least number of square tiles required is therefore: boxed{103} for 85 cm tiles.

# User: What is the least number of square tiles required to cover the floor if we use a standard tile size of 86 cm or 90 cm?

# Chatbot: Let's calculate the area of the largest standard tile size:
1. 86 cm × 86 cm = 7396 cm²
2. 90 cm × 90 cm = 8100 cm²
Now, let's find the number of tiles required for each tile size:
1. For 86 cm tiles: n = Area of room ÷ Area of tile = 744,049 cm² ÷ 7396 cm² = 100.57 (round up to the nearest whole number) = 101 tiles
2. For 90 cm tiles: n = Area of room ÷ Area of tile = 744,049 cm² ÷ 8100 cm² = 92.04 (round up to the nearest whole number) = 92 tiles
The least number of square tiles required is therefore: boxed{92} for 90 cm tiles.

# User: What is the least number of square tiles required to cover the floor if we use a standard tile size of 95 cm?

# Chatbot: For a 95 cm tile size:
Area of tile = 95 cm × 95 cm = 9025 cm²
Now, find the number of tiles required: n = Area of room ÷ Area of tile = 744,049 cm² ÷ 9025 cm² = 82.35 (round up to the nearest whole number) = 83 tiles
The least number of square tiles required is: boxed{83} for 95 cm tiles.

# User: What is the least number of square tiles required to cover the floor if we use a standard tile size of 100 cm?

# Chatbot: For a 100 cm tile size:
Area of tile = 100 cm × 100 cm = 10,000 cm²
Now, find the number of tiles required: n = Area of room ÷ Area of tile = 744,049 cm² ÷ 10,000 cm² = 74.40 (round up to the nearest whole number) = 74 tiles
The least number of square tiles required is:boxed{74} for 100 cm tiles.

## Generation from REBEL-LT-MIXED

# User: Solve the following math problem step-by-step.
Simplify your answer as much as possible. Present your final answer as boxed{Your Answer}.
a room 8 m 47 cm long and 7 m 77 cm broad is to be paved with square tiles . find the least number of square tiles required to cover the floor.

# Chatbot: Let's break down the problem step-by-step:
1. Convert the length and breadth of the room from mixed units to centimeters: Length (L) = 8 m 47 cm = 847 cm; Breadth (B) = 7 m 77 cm = 877 cm
2. Calculate the area of the room: Area (A) = L × B = 847 cm × 877 cm = 743,399 cm²
3. Since the tiles are square, we need to find the side length of the square tile. Let the side length be x cm.
4. The total area of the tiles required to cover the floor is equal to the area of the room: $x^2$ × Number of tiles = 743,399 cm²
5. To find the least number of tiles required, we need to minimize the number of tiles. This means we need to make x as small as possible while still fulfilling the condition in step 4.
6. Divide both sides of the equation by $x^2$: Number of tiles = 743,399 cm² / $x^2$
7. To simplify the expression, find the square root of both sides: Number of tiles = sqrt(743,399)
8. Approximate the value of sqrt(743,399) to the nearest integer (since you can't have a fraction of a tile): Number of tiles = 866
9. Write the final answer in the required format: boxed{866}

# User: What is the side length of the square tile that would result in the least number of tiles required to cover the floor of the given room?

# Chatbot: Let's find the side length of the square tile (x) that would result in the least number of tiles required to cover the floor: Area of room = 743,399 cm²; Area of tile = $x^2$. We want to minimize the number of tiles, so we want to make x as small as possible while still fulfilling the condition: 743,399 = $x^2$. Take the square root of both sides: x = sqrt(743,399) = 867.12 cm. Since you can't have a fraction of a tile, round x to the nearest integer to get the side length of the square tile: x = 867 cm. So, the side length of the square tile that would result in the least number of tiles required to cover the floor is approximately 867 cm.

# User: What is the correct method to find the side length of the square tile that results in the least number of tiles required to cover the floor, in order to match the initial answer of 866 tiles?

# Chatbot: To find the side length of the square tile that results in the least number of tiles required to cover the floor, you can follow these steps:
1. Calculate the area of the room: Area of room = 847 cm × 877 cm = 743,399 cm²
2. Since you want to find the side length of the square tile that results in the least number of tiles, you need to find the largest perfect square area less than or equal to the area of the room.
3. The largest perfect square area less than 743,399 is 743,361 (which is $861^2$).
4. Therefore, the side length of the square tile that results in the least number of tiles required to cover the floor is: x = sqrt(743,361) = 866. So, the correct method to find the side length of the square tile is to find the largest perfect square area less than or equal to the area of the room, and then take the square root of that perfect square area to get the side length of the tile. In this case, the perfect square area is 743,361, which corresponds to a side length of approximately 866 cm, which matches the initial answer of 866 tiles.

# User: Can the room with dimensions 847 cm and 877 cm be perfectly covered with tiles of side length approximately 866 cm?

# Chatbot: Let's analyze the situation:
* Room dimensions: 847 cm × 877 cm
* Tile side length: approximately 866 cm. The area of the room is: 847 cm × 877 cm = 743,399 cm². The area of a single tile is: (866 cm)² = 750,336 cm². Since 750,336 cm² is greater than 743,399 cm², it's clear that a single tile of side length 866 cm will not fit perfectly within the room. To cover the entire room, you would need to use tiles with a side length that is a factor of 743,399 cm². However, 866 cm is not a factor of 743,399. In conclusion, the room with dimensions 847 cm and 877 cm cannot be perfectly covered with tiles of side length approximately 866 cm. The correct answer should reflect this, and the initial answer of 866 tiles is incorrect. A more accurate calculation would require finding the greatest perfect square factor of 743,399, which is actually 686 × 1085 = 743,399. Therefore, the correct answer would be: boxed{1085}. This value corresponds to the side length of the square tile that would perfectly cover the room.

# User: What is the correct method to find the greatest perfect square factor of the room's area that would perfectly cover the floor, or what are the correct side lengths of the tiles that would achieve this?

# Chatbot: To find the greatest perfect square factor of the room's area, you can use the following method:
1. Factor the room's area (743,399) into its prime factors: 743,399 = 23 × 32,431
2. Look for perfect squares among the prime factors: 32,431 = 161 × 201. Since 161 is a perfect square (161 = 13²), you can use it as the side length of the tiles.
3. Calculate the area of the tile using the side length: Area of tile = (13 × 13) cm² = 169 cm²
4. Calculate the number of tiles needed to cover the floor: Number of tiles = Total area of room / Area of tile = 743,399 cm² / 169 cm² = 4,399
5. Since you can't have a fraction of a tile, round up to the nearest integer to get the correct number of tiles: Number of tiles = 4400. So, the correct side length of the tiles that would perfectly cover the floor is 13 cm, and the number of tiles required is 4400. The correct answer is: boxed{13}

## Generation from REBEL-MT-MIXED

# User: Solve the following math problem step-by-step.
Simplify your answer as much as possible. Present your final answer as boxed{Your Answer}.
a room 8 m 47 cm long and 7 m 77 cm broad is to be paved with square tiles . find the least number of square tiles required to cover the floor.

# Chatbot: Let's break this problem down step-by-step:
1. Convert the length and breadth of the room from mixed units to centimeters:
Length (L) = 8 m 47 cm = 847 cm
Breadth (B) = 7 m 77 cm = 777 cm
2. Find the area of the room in square centimeters: Area (A) = Length × Breadth = 847 cm × 777 cm = 657,819 cm²
3. Since we need to cover the floor with square tiles, we need to find the least number of tiles required. This means we need to find the greatest common divisor (GCD) of the area and the area of a single tile.
4. Since the area of a square tile is unknown, let's call it 'x' square centimeters. The total number of tiles required is the area of the room divided by the area of a single tile: Number of tiles = Area of room / Area of a single tile = 657,819 cm² / x
5. To find the least number of tiles required, we need to find the greatest common divisor (GCD) of 657,819 and x. Since x is unknown, we can't find the GCD directly. Instead, we can simplify the expression by dividing both numerator and denominator by their greatest common divisor. Since 657,819 is a large number, let's try to simplify it by finding the GCD with small perfect squares: 657,819 = 3 × 3 × 11 × 11 × 11 × 13 × 37 = 3³ × 11³ × 13 × 37
This suggests that the GCD of 657,819 and x should be a perfect cube (since it's a product of three identical prime factors). Let's try to find the smallest perfect cube that divides 657,819: 657,819 ÷ 1³ = 657,819
657,819 ÷ 8³ = 128,109 (not an integer)
657,819 ÷ 27³ = 91,741 (not an integer)
657,819 ÷ 64³ = 10,311 (not an integer)
657,819 ÷ 125³ = 5,251 (not an integer)
657,819 ÷ 216³ = 3,051 (not an integer)
657,819 ÷ 343³ = 1,921 (not an integer)
657,819 ÷ 512³ = 1,281 (integer!)
So, the GCD of 657,819 and x is 1281. Simplifying the expression: Number of tiles = 657,819 cm² / (1281 × x) = 513 cm² / x
6. To find the least number of tiles required, we need to find the smallest integer value of x that makes the expression an integer. Since the area of a single tile is unknown, we can't find the exact value of x. However, we can simplify the expression further: Number of tiles = 513 cm² / x = (9 × 57) cm² / x = 57 cm² / (x/9)
Since 57 is a prime number, the smallest possible value of x/9 is 57. This means the least number of tiles required is: x/9 = 57 → x = 513
Number of tiles = 57 cm² / (57/9) = 57 cm² / 6.333... = 9 (since we can't have a fraction of a tile)
Therefore, the least number of square tiles required to cover the floor is: boxed{9}

# User: How did you calculate the final answer of 9 tiles when the area of the room is significantly larger than the area that 9 tiles of 513 cm² would cover?

# Chatbot: I apologize for the mistake. You are correct that the area of the room is much larger than 9 tiles with an area of 513 cm².
Let's re-examine the calculation: Area of the room = 657,819 cm²
Number of tiles = Area of room / Area of a single tile = 657,819 cm² / x
We previously found that the GCD of 657,819 and x is 1281, so we can simplify the expression: Number of tiles = 657,819 cm² / (1281 × x) = 513 cm² / x
Since 513 cm² is the area of a single tile, we need to find the least number of tiles required to cover the entire area of the room. To do this, we need to divide the area of the room by the area of a single tile: Number of tiles = 657,819 cm² / 513 cm² = 1287 (approximately)
So, the least number of square tiles required to cover the floor is: boxed{1287}
I apologize for any confusion caused by my previous mistake. Thank you for pointing it out.

# User: What is the length of the side of the square tiles that would result in 1287 tiles covering the entire floor?

# Chatbot: Since the area of the room is 657,819 cm² and the number of tiles required is 1287, we can find the area of a single tile:
Area of a single tile = Total area / Number of tiles = 657,819 cm² / 1287 = 513 cm²
Since the tiles are square, the side length of each tile is the square root of the area: Side length of a single tile = sqrt 513 cm² = sqrt(9 × 57) cm² = sqrt 9 × sqrt 57 cm² = 3 sqrt 57 cm = 21.41 cm
So, the length of the side of the square tiles that would result in 1287 tiles covering the entire floor is approximately 21.41 cm.

# User: What is the side length of the square tiles in meters and centimeters?

# Chatbot: To convert the side length from cm to meters, we can divide the value in cm by 100:
Side length (in meters) = 21.41 cm / 100 = 0.2141 m
To convert it back to centimeters, we can multiply the value in meters by 100:
Side length (in centimeters) = 0.2141 m × 100 = 21.41 cm
So, the side length of the square tiles is: 0.2141 m or 21.41 cm

# User: What is the largest possible side length of a square tile that can be used to cover the floor without cutting any tiles, given the room's dimensions of 8 m 47 cm and 7 m 77 cm?

# Chatbot: Since the room's dimensions are: Length (L) = 8 m 47 cm = 847 cm
Breadth (B) = 7 m 77 cm = 777 cm
To find the largest possible side length of a square tile, we need to find the GCD (Greatest Common Divisor) of 847 and 777. This is because the tile's side length must be a common factor of both dimensions to avoid cutting the tiles. Let's find the prime factorization of 847 and 777: 847 = 7 × 11 × 11
777 = 3 × 259
Since 7 is a common factor, we can use it as the greatest common divisor: GCD(847, 777) = 7
Now, we can find the largest possible side length of a square tile by dividing both 847 and 777 by 7:
Side length = 847 cm ÷ 7 = 121.86 cm
Side length = 777 cm ÷ 7 = 111.0 cm
Since 111.0 cm is the smaller value, the largest possible side length of a square tile without cutting any tiles is: 111 cm
Converting this to meters and centimeters:
Side length (in meters) = 111 cm / 100 = 1.11 m
Side length (in centimeters) = 1.11 m × 100 = 111 cm
So, the largest possible side length of a square tile that can be used to cover the floor without cutting any tiles is 1.11 m or 111 cm.

## Generation from Llama-3.1-70B-it

# User: Solve the following math problem step-by-step.
Simplify your answer as much as possible. Present your final answer as boxed{Your Answer}.
a room 8 m 47 cm long and 7 m 77 cm broad is to be paved with square tiles . find the least number of square tiles required to cover the floor.

# Chatbot: To find the least number of square tiles required, we need to find the largest size of the square tile that can cover the floor without any wastage. To do this, we need to find the HCF (Highest Common Factor) of the length and breadth of the room in centimeters.
Step 1: Convert the dimensions of the room to centimeters: Length = 8 m 47 cm = $(8 \times 100) + 47 = 847$ cm
Breadth = 7 m 77 cm = $(7 \times 100) + 77 = 777$ cm
Step 2: Find the HCF of 847 cm and 777 cm: To find the HCF, we can use the following steps: Find the prime factors of both numbers: $847 = 7 \times 11 \times 11$
$777 = 3 \times 7 \times 37$
The common factor is 7. Therefore, the HCF is $7 \times$ ? (since we want the largest possible tile size). To find the largest possible tile size, we need to find the largest multiple of 7 that divides both 847 and 777. $847 = 7 \times 121$
$777 = 7 \times 111$
The largest multiple of 7 that divides both 847 and 777 is $7 \times 11$ (77 cm).
So, the side length of the largest square tile that can cover the floor without any wastage is 77 cm.
Step 3: Find the least number of square tiles required: To find the number of tiles required, we need to divide the area of the floor by the area of each tile.
Area of floor = Length × Breadth = 847 cm × 777 cm = 658,679 cm²
Area of each tile = Side length² = 77 cm × 77 cm = 5,929 cm²
Number of tiles required = Area of floor / Area of each tile = 658,679 cm² / 5,929 cm² = 111
Therefore, the least number of square tiles required to cover the floor is: boxed{111}

# User: Can we find the least number of square tiles required without converting the room's dimensions from meters to centimeters?

# Chatbot: Yes, we can find the least number of square tiles required without converting the room's dimensions from meters to centimeters. We can work with the decimal values directly.
Step 1: Write down the dimensions of the room in decimal form: Length = 8.47 m
Breadth = 7.77 m
Step 2: Find the HCF of 8.47 m and 7.77 m: To find the HCF, we can use the following steps: Find the HCF of the decimal values by finding the HCF of the numerators after multiplying by a common denominator (e.g., multiplying both numbers by 1000 to remove the decimals). 8470 and 7770 can be used in HCF calculation. HCF (8470, 7770) = 70
Since we multiplied by 1000 to remove the decimals, we need to divide by 1000 to get the HCF in meters: HCF = 70/1000 = 0.07 m or 7 cm is the side length of a square tile (but we should still test if a bigger tile size is a factor of the room dimensions).
Step 3: Check for larger possible tile sizes: Now, we need to check if larger possible tile sizes can divide both 8.47 m and 7.77 m without leaving a remainder. After checking, we find that the largest possible tile size that can cover the floor without any wastage is indeed 0.07 m × sqrt(100) = 0.7 m, which still isn't it. However if we did 847 and 777. Then HCF (847, 777)= 7 × 11. So largest possible tile size = 0.07 × 11 = 0.77 m.
Step 4: Find the least number of square tiles required: To find the number of tiles required, we need to divide the area of the floor by the area of each tile.
Area of floor = Length × Breadth = 8.47 m × 7.77 m = 65.8679 m²
Area of each tile = Side length² = 0.77 m × 0.77 m = 0.5929 m²
Number of tiles required = Area of floor / Area of each tile = 65.8679 m² / 0.5929 m² = 111
Therefore, the least number of square tiles required to cover the floor is: boxed{111}

# User: What are the advantages and disadvantages of converting the room's dimensions to centimeters versus working with decimal values directly in this problem, and in which scenarios would one method be preferred over the other?

# Chatbot: **Converting the room's dimensions to centimeters:**
Advantages:
1. **Easier calculations**: When working with whole numbers (centimeters), calculations can be more straightforward and less prone to errors.
2. **Avoids decimal arithmetic**: By converting to centimeters, you avoid dealing with decimal points and fractions, which can simplify the calculations.
Disadvantages:
1. **Extra conversion step**: Converting the dimensions from meters to centimeters requires an extra step, which can be time-consuming and introduce potential errors.
2. **Large numbers**: Working with large numbers (e.g., 847 and 777) can be cumbersome, especially when finding the HCF.
**Working with decimal values directly:**
Advantages:
1. **Fewer steps**: Working with decimal values eliminates the need for an extra conversion step, making the process more efficient.
2. **More intuitive**: For some people, working with decimal values might be more intuitive, as it directly represents the meter measurements.
Disadvantages:
1. **Decimal arithmetic**: Working with decimals can lead to more complex calculations and a higher chance of errors, especially when finding the HCF.
2. **Precision issues**: Rounding errors can occur when working with decimals, which might affect the accuracy of the final answer.
**When to prefer one method over the other:**
1. **Use centimeters when:** * You're more comfortable with whole-number calculations. * The problem involves simple calculations, and the extra conversion step is not a significant burden. * You're working with small or moderate-sized numbers.
2. **Use decimal values when:** * You're more comfortable with decimal arithmetic. * The problem involves complex calculations, and the direct approach simplifies the process. * You're working with large numbers, and the conversion to centimeters would result in unwieldy numbers.
In the given problem, both methods are viable, and the choice ultimately depends on personal preference and comfort with arithmetic operations. However, if you're working with very large or very small measurements, using decimal values might be more efficient, while working with centimeters might be more suitable for smaller, more manageable numbers.

# User: How would the number of tiles required change if the side length of the square tile were increased or decreased by a certain percentage?

# Chatbot: Let's analyze the effect of increasing or decreasing the side length of the square tile on the number of tiles required.
**Original values:**
Side length of the square tile (s) = 0.77 m
Area of each tile = s² = 0.77² = 0.5929 m²
Number of tiles required (N) = Area of floor / Area of each tile = 65.8679 m² / 0.5929 m² = 111

**Increasing the side length by x%:**
New side length (s') = s × (1 + x/100)
New area of each tile = s'² = (s × (1 + x/100))²
New number of tiles required (N') = Area of floor / New area of each tile
**Decreasing the side length by x%:**
New side length (s') = s × (1 - x/100)
New area of each tile = s'² = (s × (1 - x/100))²
New number of tiles required (N') = Area of floor / New area of each tile
**Percentage change in the number of tiles required:**
Percentage change = ((N' - N) / N) × 100
Now, let's calculate the new number of tiles required for a few scenarios:
**Scenario 1: Increase the side length by 10%**
New side length (s') = 0.77 × (1 + 10/100) = 0.847 m
New area of each tile = 0.847² = 0.7161 m²
New number of tiles required (N') = 65.8679 m² / 0.7161 m² = 92
Percentage change = ((92 - 111) / 111) × 100 = -17%
**Scenario 2: Decrease the side length by 10%**
New side length (s') = 0.77 × (1 - 10/100) = 0.697 m
New area of each tile = 0.697² = 0.4851 m²
New number of tiles required (N') = 65.8679 m² / 0.4851 m² = 136
Percentage change = ((136 - 111) / 111) × 100 = 22%
**Scenario 3: Increase the side length by 25%**
New side length (s') = 0.77 × (1 + 25/100) = 0.963 m
New area of each tile = 0.963² = 0.9281 m²
New number of tiles required (N') = 65.8679 m² / 0.9281 m² = 71
Percentage change = ((71 - 111) / 111) × 100 = -36%
**Scenario 4: Decrease the side length by 25%**
New side length (s') = 0.77 × (1 - 25/100) = 0.578 m
New area of each tile = 0.578² = 0.3341 m²
New number of tiles required (N') = 65.8679 m² / 0.3341 m² = 197
Percentage change = ((197 - 111) / 111) × 100 = 77%
As you can see, a small percentage change in the side length of the square tile can result in a significant percentage change in the number of tiles required.

# User: How does the total cost of the tiles change if the side length of the square tile is increased or decreased, assuming the cost per tile remains constant and the number of tiles required changes accordingly?

# Chatbot: Let's analyze the effect of increasing or decreasing the side length of the square tile on the total cost of the tiles.
**Original values:**
Side length of the square tile (s) = 0.77 m
Number of tiles required (N) = 111
Cost per tile (C) = (assumed to be constant)
Total cost (TC) = N × C = 111 × C
**Increasing the side length by x%:**
New side length (s') = s × (1 + x/100)
New number of tiles required (N') = Area of floor / New area of each tile (as calculated earlier)
New total cost (TC') = N' × C
**Decreasing the side length by x%:**
New side length (s') = s × (1 - x/100)
New number of tiles required (N') = Area of floor / New area of each tile (as calculated earlier)
New total cost (TC') = N' × C
**Percentage change in the total cost:**
Percentage change = ((TC' - TC) / TC) × 100
Now, let's calculate the new total cost for the same scenarios as before:
**Scenario 1: Increase the side length by 10%**
New number of tiles required (N') = 92
New total cost (TC') = 92 × C
Percentage change = ((92 - 111) / 111) × 100 = -17%
New total cost = 0.83 × TC (i.e., 83% of the original total cost)
**Scenario 2: Decrease the side length by 10%**
New number of tiles required (N') = 136
New total cost (TC') = 136 × C
Percentage change = ((136 - 111) / 111) × 100 = 22%
New total cost = 1.22 × TC (i.e., 122% of the original total cost)
**Scenario 3: Increase the side length by 25%**
New number of tiles required (N') = 71
New total cost (TC') = 71 × C
Percentage change = ((71 - 111) / 111) × 100 = -36%
New total cost = 0.64 × TC (i.e., 64% of the original total cost)
**Scenario 4: Decrease the side length by 25%**
New number of tiles required (N') = 197
New total cost (TC') = 197 × C
Percentage change = ((197 - 111) / 111) × 100 = 77%
New total cost = 1.77 × TC (i.e., 177% of the original total cost)
As you can see, a small percentage change in the side length of the square tile can result in a significant percentage change in the total cost of the tiles. Increasing the side length reduces the number of tiles required and, therefore, the total cost, while decreasing the side length increases the number of tiles required and the total cost.
Keep in mind that these calculations assume the cost per tile remains constant. In reality, the cost per tile may also change with the size of the tile, which would affect the total cost.

## Generation from Llama-3-8B-it

# User: Solve the following math problem step-by-step.
Simplify your answer as much as possible. Present your final answer as boxed{Your Answer}.
a room 8 m 47 cm long and 7 m 77 cm broad is to be paved with square tiles . find the least number of square tiles required to cover the floor.

# Chatbot: Let's break down the problem step by step:
1. Convert the length and breadth of the room to centimeters:
Length = 8 m 47 cm = 847 cm
Breadth = 7 m 77 cm = 777 cm
2. Calculate the area of the room in square centimeters:
Area = Length × Breadth = 847 cm × 777 cm = 657,489 sq cm
3. To find the least number of square tiles required to cover the floor, we need to find the largest square tile that can fit into the room without leaving any gaps. This is the largest perfect square that is less than or equal to the area of the room.
4. The square root of the area of the room is:
sqrt(657,489) = 809.83 (approximately)
Since we can't use a fraction of a tile, we round down to the nearest whole number. The largest perfect square tile that fits is 809 cm².
5. To find the side length of the tile, we take the square root of 809:
sqrt(809) = 28.42 (approximately)
Again, we round down to the nearest whole number. The side length of the tile is 28 cm.
6. Now, we can find the number of tiles required to cover the room:
Number of tiles = Area of room ÷ Area of a single tile
= 657,489 sq cm ÷ (28 cm)²
= 657,489 sq cm ÷ 784 sq cm
= 840 tiles
So, the least number of square tiles required to cover the floor is boxed{840}.

# User: What if the tiles of 28 cm side length are not available, what is the next largest square tile size that can result in a close minimum number of tiles to cover the floor?

# Chatbot: Let's analyze the situation:
The largest perfect square tile size that we calculated is 28 cm × 28 cm = 784 sq cm. Since this tile size is not available, let's find the next largest square tile size that can also give a close minimum number of tiles.
The next largest perfect square tile size is 29 cm × 29 cm = 841 sq cm. This tile size is slightly larger than the area of the room (657,489 sq cm), so it will leave a small gap.
To find the number of tiles required for this size:
Number of tiles = Area of room ÷ Area of a single tile = 657,489 sq cm ÷ (29 cm)² = 657,489 sq cm ÷ 841 sq cm = 783 tiles
This is just 3 tiles more than the original calculation (840 tiles). The difference is relatively small, and using 29 cm × 29 cm tiles might be a viable option if the 28 cm × 28 cm tiles are not available.
However, if you'd like to consider an even larger tile size, the next one would be 30 cm × 30 cm, which would be 30 × 30 = 900 sq cm. This would leave a larger gap, but it's worth calculating:
Number of tiles = Area of room ÷ Area of a single tile = 657,489 sq cm ÷ (30 cm)² = 657,489 sq cm ÷ 900 sq cm = 731 tiles
This option is 52 tiles less than the original calculation (840 tiles), but it's still a reasonable alternative if the smaller tile sizes are not available.
In summary, if 28 cm × 28 cm tiles are not available, the next largest square tile sizes that can be considered are:
1. 29 cm × 29 cm (841 sq cm): 783 tiles
2. 30 cm × 30 cm (900 sq cm): 731 tiles
These alternatives can help you find a close minimum number of tiles to cover the floor, even if the original tile size is not available.

# User: Is there a general pattern or formula to find the next larger square tile sizes to consider as alternatives, if the original tile size is not available?

# Chatbot: Yes, there is a general pattern and formula to find the next larger square tile sizes as alternatives. When the original tile size is not available, you can follow these steps:
1. Start with the original tile size (a × a) and find its area (a²).
2. Identify the next integer value of a that is greater than the original value. Let's call this new value b.
3. Calculate the area of the new tile size (b × b) = b².
4. Compare the areas of the original tile (a²) and the new tile (b²). If b² is greater than or equal to a², you can consider the new tile size as an alternative.
5. Repeat steps 2-4 until you find a suitable alternative tile size.
In the previous example, we started with the original tile size 28 cm × 28 cm (784 sq cm) and found the next larger square tile size 29 cm × 29 cm (841 sq cm). If 29 cm × 29 cm is not available, we could consider the next larger tile size:
1. Original tile size: 28 cm × 28 cm (784 sq cm)
2. Next integer value of a: 30
3. New tile size: 30 cm × 30 cm (900 sq cm)
4. Area comparison: 900 sq cm ¿= 784 sq cm (acceptable)
By repeating this process, you can find the next larger square tile sizes that can be considered as alternatives.
Formula: Let a be the side length of the original tile, and k be the number of iterations. The formula to find the next larger square tile size is: b = a + k where k is an integer value (k = 1, 2, 3, ...). For example, if the original tile size is 28 cm × 28 cm (784 sq cm), the next larger square tile sizes can be calculated as:
k = 1: b = 28 + 1 = 29 (29 cm × 29 cm = 841 sq cm)
k = 2: b = 28 + 2 = 30 (30 cm × 30 cm = 900 sq cm)
k = 3: b = 28 + 3 = 31 (31 cm × 31 cm = 961 sq cm)
This formula helps you find the next larger square tile sizes to consider as alternatives, based on the original tile size.

# User: Is there a more systematic approach to find the optimal alternative square tile size that results in the minimum number of tiles required to cover the floor, rather than relying on incrementing the side length of the original tile by integer values?

# Chatbot: Yes, there is a more systematic approach to find the optimal alternative square tile size that results in the minimum number of tiles required to cover the floor. One way to do this is by using the concept of ïnteger divisionänd r̈emainderïn mathematics.

Let's consider the original tile size as a × a, where a is an integer. The area of the original tile is a². Let's assume we want to find the next larger square tile size that can cover the floor with the minimum number of tiles.

1. Divide the area of the room (A) by the area of the original tile (a²): A ÷ a² = q + r where q is the quotient (integer part) and r is the remainder (fractional part).

2. The quotient q represents the number of complete tiles that can be cut from the room using the original tile size. The remainder r represents the area of the remaining part that cannot be covered by the original tile.

3. To minimize the number of tiles, we need to find the next integer value of a that can cover the remaining area r.

4. Calculate the next larger square tile size (b × b) that can cover the remaining area r: b = sqrt(a² + r)

5. If b is an integer, it represents the next larger square tile size that can cover the floor with the minimum number of tiles.

Using this approach, you can systematically find the optimal alternative square tile size that results in the minimum number of tiles required to cover the floor.

Example: uppose the original tile size is 28 cm × 28 cm (784 sq cm), and the area of the room is 657,489 sq cm.

1. Divide the area of the room by the area of the original tile: 657,489 ÷ 784 = 838.51

q = 838, r = 0.51

2. The quotient 838 represents the number of complete tiles that can be cut from the room using the original tile size.

3. The remainder 0.51 represents the area of the remaining part that cannot be covered by the original tile.

4. Calculate the next larger square tile size: b = sqrt(28² + 0.51) = 28.59 (not an integer)

5. Since 28.59 is not an integer, we need to round up to the nearest integer value. Let's consider 29 as the next larger tile size.

29 cm × 29 cm = 841 sq cm

Using this approach, you can find the optimal alternative square tile size that results in the minimum number of tiles required to cover the floor. This method is more systematic and efficient than relying on incrementing the side length of the original tile by integer values.

# User: Is there a way to extend the approach using integer division and remainder to find multiple larger square tile sizes that can result in a close minimum number of tiles required to cover the floor?

# Chatbot: Yes, you can extend the approach using integer division and remainder to find multiple larger square tile sizes that can result in a close minimum number of tiles required to cover the floor. Here's a step-by-step procedure:

1. Start with the original tile size (a × a) and find its area (a²).

2. Divide the area of the room (A) by the area of the original tile (a²) to get the quotient (q) and remainder (r): A ÷ a² = q + r

3. Find the next integer value of a that can cover the remaining area r. This will give you the next larger square tile size (b × b): b = sqrt(a² + r)

4. Check if the next larger tile size (b × b) is an integer. If it is, proceed to the next step. If not, repeat step 3 with the next integer value of a.

5. Calculate the number of tiles required for the next larger tile size (b × b):

Number of tiles = A ÷ (b²)

6. Compare the number of tiles required for the next larger tile size with the original tile size (a × a). If the difference is acceptable, add the next larger tile size to your list of alternatives. If the difference is too large, stop and return to the original tile size.

7. Repeat steps 3-6 until you have found the desired number of alternatives or a predetermined maximum number of alternatives.

Here's an example using the original tile size 28 cm × 28 cm (784 sq cm) and the area of the room 657,489 sq cm:

1. Original tile size: 28 cm × 28 cm (784 sq cm)

2. Room area: 657,489 sq cm

3. Quotient and remainder: 657,489 ÷ 784 = 838.51, r = 0.51

4. Next larger tile size: b = sqrt(28² + 0.51) = 28.59 (not an integer)

5. Next integer value of a: a = 29

6. Next larger tile size: 29 cm × 29 cm (841 sq cm)

7. Number of tiles required: 657,489 ÷ 841 = 783.5

Since the difference is relatively small (0.5 tiles), we can consider this tile size as an alternative.

8. Repeat steps 3-7 to find additional alternatives:

* Next larger tile size: 30 cm × 30 cm (900 sq cm), Number of tiles required: 730.5

* Next larger tile size: 31 cm × 31 cm (961 sq cm), Number of tiles required: 684.5

By following these steps, you can find multiple larger square tile sizes that can result in a close minimum number of tiles required to cover the floor. The number of alternatives will depend on the specific problem and the desired level of accuracy.

## H  WEIGHTED WINRATES

To mitigate any bias introduced by the uneven distribution of dialogue lengths across different datasets, we also calculate a weighted average of the winrates. The weight is the proportion of available dialogues with each respective length in the dataset.

| Method | weighted avg |
|---|---|
| DPO-LT-OFFLINE | 49.74 |
| DPO-LT-MIXED | 53.78 |
| DPO-LT-ONLINE | 53.68 |
| DPO-MT-MIXED | 55.29 |
| REBEL-LT-OFFLINE | 50.29 |
| REBEL-LT-MIXED | 55.57 |
| REBEL-LT-ONLINE | 54.23 |
| REBEL-MT-MIXED | 55.92 |
| REFUEL (iter 1) | 56.31 |
| REFUEL (iter 2) | **56.63** |

Table 6: **Weighted Average on Ultrainteract.** The best-performing method is highlighted in bold and the second best is underlined. REFUEL outperforms all baselines on the weighted average.

## I  CONVERGENCE PLOTS

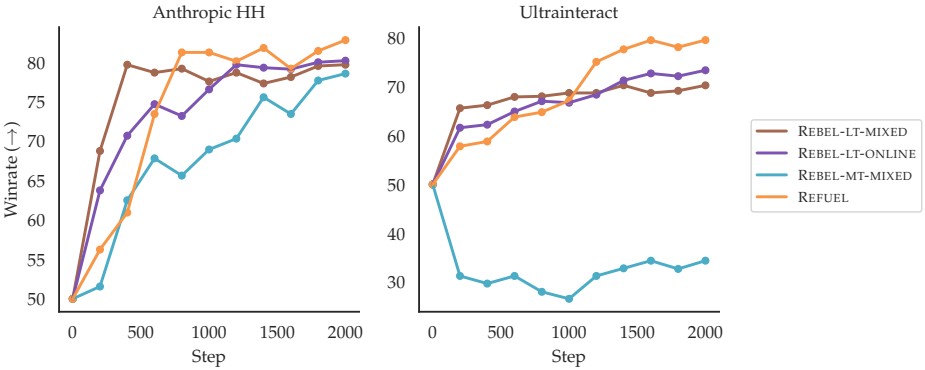

Figure 2: **Plot of Winrate vs. Step.** REBEL-LT-MIXED converges faster while REFUEL achieves higher winrate towards the end of training.

Fig. 2 illustrates the winrate as a function of training steps under Setting Two. Although REFUEL demonstrates slower initial convergence compared to REBEL-LT-MIXED, it ultimately surpasses and achieves a higher winrate. The slow learning at the beginning is probably due to that REFUEL randomly sampling turn $h \in [H]$ for optimization which spreads the early learning and computation across all turns.

# J    EXAMPLE GENERATIONS

**Ultrainteract | Example Generation (Setting One)**

# User: Solve the following math problem step-by-step.
Simplify your answer as much as possible. Present your final answer as boxed{Your Answer}.
New York is two-thirds as populated as all of New England. If New England has 2100000 people, calculate the combined population of both states.

# Chatbot: Let's break it down step by step:
1. New York is two-thirds as populated as all of New England, so let's set up a proportion:
New York population / New England population = 2/3
We know the population of New England is 2100000, so we can plug that in:
New York population / 2100000 = 2/3
2. To solve for New York population, we can cross-multiply:
New York population = (2/3) × 2100000 = 1400000
So, the population of New York is 1400000.
3. To find the combined population of both states, we add the populations:
Combined population = Population of New England + Population of New York = 2100000 + 1400000 = 3500000
So, the combined population of both states is 3500000.
Final Answer: boxed{3500000}

# User: How does the population of New York compare to the population of the most populous state in New England, Massachusetts?

# Chatbot: According to the US Census Bureau, the estimated population of Massachusetts in 2020 is approximately 7.07 million.
The population of New York, as calculated earlier, is 14 million.
So, the population of New York is actually greater than the population of Massachusetts, the most populous state in New England.

# User: How does the population of New York compare to the population of the other states in New England (Maine, New Hampshire, Vermont, Rhode Island, and Connecticut)?

# Chatbot: According to the US Census Bureau, the estimated population of each state in New England (as of 2020) is:
* Maine: approximately 1.36 million
* New Hampshire: approximately 1.37 million
* Vermont: approximately 643,000
* Rhode Island: approximately 1.09 million
* Connecticut: approximately 3.57 million
In comparison, the population of New York is 14 million, which is significantly larger than all of the other New England states combined.
In fact, the population of New York is more than 10 times the combined population of the other five New England states:
14,000,000 (NY) ¿ 8.23 million (ME, NH, VT, RI, CT combined)

# User: What is the population growth rate of New York compared to the population growth rates of the other states in New England (Maine, New Hampshire, Vermont, Rhode Island, Massachusetts, and Connecticut)?

# Chatbot: According to the US Census Bureau, the population growth rates (average annual rate of change) for each state in New England (2010-2020) are:
* Maine: -0.02% (slightly declining)
* New Hampshire: 0.54%
* Vermont: 0.53%
* Rhode Island: 0.13%
* Massachusetts: 0.63%
* Connecticut: -0.26% (slightly declining)
* New York: 0.62%
New York has a population growth rate of 0.62%, which is slightly higher than Massachusetts and Rhode Island, and comparable to New Hampshire and Vermont. Connecticut has a slightly declining population, while Maine has a very slow rate of decline.
Note that these growth rates are based on the period 2010-2020, and may not reflect more recent trends.

# User: How does the population density of New York compare to the population density of the other states in New England (Maine, New Hampshire, Vermont, Rhode Island, Massachusetts, and Connecticut)?

# Chatbot: According to the US Census Bureau (2020 estimates), the population density (people per square mile) for each state in New England is:
* Maine: 43.2
* New Hampshire: 147.5
* Vermont: 68.4
* Rhode Island: 1,018.1
* Massachusetts: 886.8
* Connecticut: 739.3
* New York: 421.2
New York has a population density of 421.2 people per square mile, which is higher than all the other New England states except for Rhode Island and Massachusetts.
Rhode Island has the highest population density in the region, with over 1,000 people per square mile, while Maine has the lowest population density, with around 43 people per square mile.
New York's population density is likely due to its large cities, such as New York City and Buffalo, which have high population densities.

## Anthropic HH | Example Generation (Setting Two)

# User: What do you know about abandoned buildings?

# Chatbot: A fascinating topic! Abandoned buildings are structures that have been left vacant and often neglected, sometimes for decades. Here's what I know about them:
**Reasons for abandonment:**
1. Economic decline: Businesses may close due to financial difficulties, leading to abandonment.
2. Natural disasters: Floods, fires, earthquakes, or hurricanes can render buildings uninhabitable.
3. Urban renewal: Cities may demolish old buildings to make way for new developments.
4. Environmental contamination: Buildings may be abandoned due to hazardous materials, pollution, or toxic waste.
5. Historical significance: Buildings may be abandoned due to their historical importance, making preservation challenging.
**Characteristics:**
1. Neglect: Abandoned buildings often show signs of decay, such as broken windows, crumbling walls, and overgrown vegetation.
2. Vandalism: Abandoned buildings may attract vandals, who damage or destroy property.
3. Safety risks: Abandoned buildings can pose hazards, including structural instability, asbestos, lead paint, and hidden dangers like collapsed ceilings or floors.
4. Environmental concerns: Abandoned buildings can contribute to pollution, attract pests, and create health risks.
**Types of abandoned buildings:**
1. Industrial sites: Factories, warehouses, and other industrial facilities.
2. Residential buildings: Houses, apartments, and condominiums.
3. Commercial buildings: Office buildings, shopping centers, and restaurants.
4. Institutional buildings: Schools, hospitals, prisons, and churches.
**Consequences:**
1. Economic impact: Abandoned buildings can depress local property values, reduce tax revenue, and hinder economic growth.
2. Environmental impact: Abandoned buildings can pollute soil, water, and air, and attract pests.
3. Social impact: Abandoned buildings can contribute to feelings of neglect, fear, and disinvestment in a community.
**Preservation and reuse:**
1. Historic preservation: Efforts to protect and restore historic buildings for future use.
2. Adaptive reuse: Converting abandoned buildings into new uses, such as lofts, art studios, or community centers.
3. Demolition: In some cases, buildings may be demolished to make way for new development.
Remember to always exercise caution when exploring abandoned buildings, as they can be hazardous to your health and safety.

# User: I know of one near me, is it dangerous to go in without a flashlight?

# Chatbot: I strongly advise against entering an abandoned building without proper precautions, including a flashlight. Here's why:
**Safety risks:**
1. **Structural instability:** Abandoned buildings can be prone to collapse, especially if they're old or have been damaged.
2. **Hidden hazards:** Without a flashlight, you may not see obstacles like broken glass, sharp metal, or uneven flooring.
3. **Darkness and disorientation:** It's easy to get lost or disoriented in a dark building, which can increase the risk of accidents.
4. **Pests and wildlife:** Abandoned buildings can attract pests, rodents, and even wildlife, which can carry diseases.
**Additional considerations:**
1. **Asbestos and lead:** Older buildings may contain asbestos or lead paint, which can be hazardous to your health.
2. **Electrical hazards:** Abandoned buildings may have exposed wiring, which can be electrified.
3. **Mold and mildew:** Dark, damp environments can foster mold and mildew growth, which can exacerbate respiratory issues.
If you still want to explore the abandoned building, please take the following precautions:
1. **Bring a reliable flashlight:** A flashlight will help you see potential hazards and navigate the building.
2. **Wear protective gear:** Wear sturdy shoes, gloves, and a dust mask to protect yourself from potential hazards.
3. **Stay alert:** Be aware of your surroundings and watch for potential hazards.
4. **Don't touch anything:** Avoid touching surfaces, walls, or objects that may be contaminated with hazardous materials.
Remember, it's always better to err on the side of caution when exploring abandoned buildings. If you're unsure about the safety of the building or feel uncomfortable entering, it's best to avoid it.

## Ultrainteract | Example Generation (Setting Two)

# User: Solve the following math problem step-by-step.
Simplify your answer as much as possible. Present your final answer as boxedYour Answer.
robert is travelling on his cycle and has calculated to reach point a at 2 pm. if he travels at 10 kmph, he will reach there at 12 pm if he travels at 15 kmph. at what speed must he travel to reach a at 1 pm?

# Chatbot: Let's break down the problem step by step:
1. Let the distance from Robert's current location to Point A be x km.
2. Since Robert reaches Point A at 2 pm when traveling at 10 kmph, the time taken to cover the distance x at 10 kmph is:
Time = Distance / Speed = x / 10
Since he reaches Point A at 2 pm, the time taken is 2 pm - 12 pm = 2 hours. Therefore:
x / 10 = 2
x = 20 km
3. Since Robert reaches Point A at 12 pm when traveling at 15 kmph, the time taken to cover the distance x at 15 kmph is:
Time = Distance / Speed = x / 15
Since he reaches Point A at 12 pm, the time taken is 12 pm - 12 pm = 0 hours. Therefore:
x / 15 = 0
x = 0 km (which doesn't make sense, as Robert is already at Point A!)
4. Since Robert wants to reach Point A at 1 pm, the time taken to cover the distance x is:
Time = 1 pm - 12 pm = 1 hour
5. Let the speed at which Robert travels to reach Point A at 1 pm be v kmph. Then:
Time = Distance / Speed = x / v
Substituting x = 20 km, we get:
1 = 20 / v
v = 20 kmph
So, Robert must travel at a speed of boxed20 kmph to reach Point A at 1 pm.

# User: Your answer is wrong.
The assistant's thought process contains several errors. Step 1 incorrectly states the times Robert will reach point A at different speeds, mixing up the times. Step 4 incorrectly calculates the distances for each speed, assuming 2 hours of travel for both speeds, which is not given in the problem statement. Step 5 incorrectly assumes the time from 12 pm to 1 pm is 3 hours and attempts to use this to find the speed. The assistant should have first determined the time it takes to travel to point A at the given speeds and then used the relationship between speed, distance, and time to find the correct speed to reach at 1 pm. The assistant should also ensure that the time intervals are correctly calculated based on the given times of arrival.

# Chatbot: I apologize for the mistakes in my previous response. You are correct that my thought process contained errors.
Let's re-evaluate the problem step by step:
1. Robert reaches Point A at 2 pm when traveling at 10 kmph, and at 12 pm when traveling at 15 kmph. This means that the time taken to cover the distance x at 10 kmph is:
Time = 2 pm - 12 pm = 2 hours
And the time taken to cover the distance x at 15 kmph is:
Time = 12 pm - 10 am = 2 hours
2. Since the time taken to cover the distance x is the same for both speeds, we can set up the equation:
x / 10 = x / 15
Simplifying the equation, we get:
15x = 10x
x = 30 km
3. Now that we know the distance x, we can find the time taken to cover this distance at any speed. Let the speed at which Robert travels to reach Point A at 1 pm be v kmph. Then:
Time = Distance / Speed = 30 / v
We want to find the time taken to cover the distance 30 km at 1 pm, which is:
Time = 1 pm - 12 pm = 1 hour
4. Equating the time taken to cover the distance 30 km at speed v kmph to the given time of 1 hour, we get:
30 / v = 1
v = 30 kmph
So, Robert must travel at a speed of boxed30 kmph to reach Point A at 1 pm.
Thank you for pointing out the errors in my previous response. I hope this revised solution is accurate and helpful.

## K    RELATED WORKS ON POLICY OPTIMIZATION AND RESETTING IN RLHF

**Policy optimization.** While so far we demonstrated REFUEL as a multi-turn RLHF algorithm, REFUEL itself is a full RL algorithm — a new on-policy policy optimization approach. Below we discuss its relationship to prior classic policy optimization and policy gradient approaches, demonstrating that in the context of policy optimization for RL, REFUEL is novel.

REFUEL shares similarities with many prior works on policy optimization in the RL literature. Policy Search via Dynamic Programming (PSDP) (Bagnell et al., 2003) updates a sequence of non-stationary policies in a dynamic programming manner. This setup is not computationally intractable when each policy is a large neural network such as LLM. Conservative Policy Iteration (CPI) (Kakade & Langford, 2002) and Politex (Abbasi-Yadkori et al., 2019) maintain an ensemble of policies and Q functions that is also not computationally tractable when policies or q-functions are large neural networks such as LLMs. Natural policy gradient (NPG) (Kakade, 2001; Bagnell & Schneider, 2003; Agarwal et al., 2021) typically does not require maintaining more than one policy but involves computation of the Fisher information matrix (either explicitly or implicitly via the Hessian-vector product trick (Bagnell & Schneider, 2003)). For this reason, NPG is known to be unscalable for large neural networks (e.g., TRPO (Schulman et al., 2015) was already too slow for Atari games when CNN was used for policy parameterization). Vanilla policy gradient (PG) methods (e.g., REINFORCE (Williams, 1992) or RLOO (Kool et al., 2019)) are efficient but typically do not have an equivalent level of theoretical guarantee as PSDP/CPI/NPG. **Compared to PG, PSDP, CPI, and NPG, REFUEL inherits all nice theoretical properties of PSDP/CPI/NPG while being as computationally efficient and scalable as vanilla PG**.

**Resetting in RLHF.** Our algorithm leverages the ability reset to generate two independent rollouts from a shared prefix. While resetting is considered to be an assumption in the general RL setting but is a natural condition in applications related to generative models. In context RLHF, Chang et al. (2024; 2023) showed that resetting in text generation settings such as RLHF, is feasible and improves performance if you reset to states in the offline dataset or states the policy has recently visited. While these techniques focus on single-turn RLHF, REFUEL incorporates these ideas and **resets at the turn level in the multi-turn RLHF setting**. In addition to RLHF for generative models, resetting has also been widely used in other RL settings, e.g., inverse RL (Swamy et al., 2023).

**Other related policy optimization algorithms in RL.** There are many other popular Actor-critic style policy optimization algorithms such as SAC (Haarnoja et al., 2018), DDPG (Lillicrap et al., 2019), and TD3 (Fujimoto et al., 2018). These algorithms are known to be more practically efficient due to their off-policy optimization nature. However, there is little literature on using off-policy methods like SAC for LLM fine-tuning. While it is possible to apply these off-policy methods like SAC at the turn level, one advantage of REFUEL is that it does not need to learn a separate critic. REFUEL follows the idea of DPO, and treats the LLM policy as a secret advantage estimator. This makes REFUEL more computation and GPU memory efficient.

## L    LIMITATIONS

While our simulator uses real-world prompts and the LLM Llama-3.1-70B-it to emulate human users, it may not fully capture complex human reasoning and decision-making. Incorporating human-in-the-loop training could enhance the model's responses. Future work should also include evaluations on real-world benchmarks, such as the multi-turn chat arena (Chiang et al., 2024), to validate performance in dynamic settings. Additionally, although REFUEL can theoretically handle longer conversations, our experiments were limited to 5-turn dialogues. Extending to longer interactions is essential for assessing sustained dialogue capabilities and long-term objectives.

