# OpenReview forum: "Regressing the Relative Future: Efficient Policy Optimization for Multi-turn RLHF"
_ICLR.cc/2025/Conference — ICLR 2025 Poster_

### Official Review · Reviewer_GJRW · 2024-11-03

**Soundness:** 3
**Presentation:** 3
**Contribution:** 3
**Rating:** 8
**Confidence:** 3

**Summary:**

This paper introduces REFUEL (REgressing the RElative FUturE for reinforcement Learning), a novel approach for multi-turn Reinforcement Learning from Human Feedback (RLHF) in Large Language Models (LLMs). The key contributions include:
1. A new algorithm that addresses covariate shift in multi-turn dialogue by using self-generated data and treating the task as a sequence of regression problems
2. Theoretical guarantees showing REFUEL can match any policy covered by the training set
3. Strong empirical performance, where an 8B parameter model fine-tuned with REFUEL outperforms a 70B parameter model on long multi-turn dialogues
4. A novel approach that eliminates the need for an explicit critic network by using a reparameterization trick and relative future rewards

**Strengths:**

1. The paper addresses a significant limitation in current RLHF approaches by properly handling multi-turn dialogue scenarios instead of treating them as single-turn tasks.
2. The theoretical analysis is thorough and proves that REFUEL requires weaker conditions than previous methods like Natural Policy Gradient.
3. The empirical results are impressive, showing that a smaller model trained with REFUEL can outperform much larger models on long dialogues.
4. The method is computationally efficient by avoiding the need for an additional critic network.

**Weaknesses:**

1. The experimental evaluation is limited to dialogues of 5 turns or less, which may not fully demonstrate the method's capability for very long conversations.
2. The simulator used for evaluation, while sophisticated, may not fully capture real human interaction patterns.
3. The paper lacks ablation studies on different components of the algorithm and their contributions to the final performance.
4. The comparison with PPO-based methods is omitted due to computational constraints, which leaves some questions about relative performance against all potential baselines.

**Questions:**

1. Does REFUEL's approach work for other multi-turn tasks beyond dialogue, such as multi-step reasoning or sequential decision-making tasks?
2. Can you provide ablation studies showing the individual impact of key components in REFUEL? Specifically, how much does the relative future reward regression contribute versus the on-policy data generation?
3. Can you demonstrate REFUEL's effectiveness on significantly longer conversations (e.g., 10-20 turns)?

---

> ### Author Response · Authors · 2024-11-19
> **Response to Reviewer GJRW**
>
> Thank you for your valuable review of our paper. We respond to your individual questions below.
>
> > The experimental evaluation is limited to dialogues of 5 turns or less, which may not fully demonstrate the method's capability for very long conversations.
>
> > Can you demonstrate REFUEL's effectiveness on significantly longer conversations (e.g., 10-20 turns)?
>
> Thank you for the suggestion! We agree that extending the dialogue length beyond 5 turns could potentially lead to improved results. However, we decided against this for two main reasons. First, the majority of dialogues in the Ultrainteract dataset are shorter than 5 turns, so comparing against baselines like LT-OFFLINE, LT-MIXED, and MT-MIXED would not be entirely fair on dialogues more than 5 turns, as these baselines do not include sufficiently long dialogues in their offline datasets. Second, generating dialogues longer than 5 turns would be computationally prohibitive for us, as REFUEL requires multiple iterations (iter 1 and iter 2), and we rely on a large user simulation model (Llama-3.1-70B-it). The computational demands for generating dialogues of that length exceed our current capacity.
>
> > The simulator used for evaluation, while sophisticated, may not fully capture real human interaction patterns.
>
> We fully agree that the simulator cannot fully capture real human interaction. However, this simulator remains the best alternative as collecting dialogues with 5 turns for 65,000 prompts from real humans is really expensive.
>
> > The paper lacks ablation studies on different components of the algorithm and their contributions to the final performance.
>
> > Can you provide ablation studies showing the individual impact of key components in REFUEL? Specifically, how much does the relative future reward regression contribute versus the on-policy data generation?
>
> Thank you for the great suggestion. In fact, in the experiment section, the baseline methods could be viewed as the ablations you mentioned.
> - Comparing REFUEL with LT-ONLINE methods essentially isolates the contribution of relative future reward regression, as both approaches use on-policy data generation.
> - Comparing REFUEL with MT-MIXED methods isolates the contribution of on-policy data generation, as both methods use relative future reward regression.
>
> We observe that the contribution of on-policy data generation is slightly higher than that of relative future reward regression, as the LT-ONLINE methods tend to outperform MT-MIXED methods in terms of winrates.
>
> > Does REFUEL's approach work for other multi-turn tasks beyond dialogue, such as multi-step reasoning or sequential decision-making tasks?
>
> Although we didn’t apply REFUEL for multi-step reasoning tasks, we suspect that it will also work. Multi-step reasoning can be viewed as a simplified version of multi-turn dialogue, where the model autonomously performs multiple reasoning steps after an initial user query. In contrast to multi-turn dialogue, where the trajectory consists of alternating turns between the human and assistant (e.g., $\{x_1, y_1, x_2, …, y_{h-1}, x_h, y_h\}$), multi-step reasoning focuses on a single human query followed by multiple reasoning steps from the model (e.g., $\{x_1, y_1, y_2, …, y_{h-1}, y_h\}$). REFUEL could similarly be applied to this task, where the state $s_h$​ would consist of the initial query $x_1$​ and all reasoning steps up to $h$: $s_h = (x_1, y_1, …, y_{h-1})$.

---

### Official Review · Reviewer_GLhj · 2024-11-04

**Soundness:** 4
**Presentation:** 3
**Contribution:** 3
**Rating:** 6
**Confidence:** 2

**Summary:**

This work introduces REFUEL, a method addressing the covariate shift issue appearing when commonly applying single-turn RLHF methods to multi-turn dialogues. It frames the problem as a multi-step MDP and merges the two-step of actor-critic procedures into a unified procedure. It shows improvements on benchmarks such as UltraInteract and AnthropicHH when having sample conversations evaluated head-to-head by GPT4.

**Strengths:**

# Originality

To the best of my knowledge, the refuel technique and its application to multi-turn dialogue tasks is novel, as most existing work leverage an actor-critic framework.

# Quality & Clarity

The paper is well-written and easy to follow. It states clearly the limitations of current approaches and presents its contributions in a straightforward manner. The chosen hyperparameters and detailed proofs are clearly described in the appendix.

# Significance

The task of designing high-quality dialogue agents has become ever more visible in the past couple of years. Controlling generations over multiple dialogue turns is critical to the success of such agents, as it could enable more powerful and human-like capabilities with dialogues that carry an extended amount of context and information in previous turns and that are strategically generated in order to achieve a macroscopic goal. The paper is making a significant stride towards that.

**Weaknesses:**

The evaluations proposed could be more extensive. In particular, it may be interesting to show evaluations on more common single-turn dialogue tasks (e.g. QA tasks) in order to show whether the quality is degraded for these tasks. It is also surprising to see a strong degradation in quality for turns #0 and #1 compared to the raw Llama 8B model, and the paper does not provide an explanation for this.

**Questions:**

* Why are dialogues filtered to a maximum of 5 turns? It would seem that the performance improvement would be larger on longer dialogues.
* Why is performance degraded over most alternative setups for turns #0 and #1?

---

> ### Author Response · Authors · 2024-11-19
> **Response to Reviewer GLhj**
>
> Thank you for your valuable review of our paper. We respond to your individual questions below.
>
> > The evaluations proposed could be more extensive. In particular, it may be interesting to show evaluations on more common single-turn dialogue tasks (e.g. QA tasks) in order to show whether the quality is degraded for these tasks. It is also surprising to see a strong degradation in quality for turns #0 and #1 compared to the raw Llama 8B model, and the paper does not provide an explanation for this.
>
> > Why is performance degraded over most alternative setups for turns #0 and #1?
>
> If we are only doing single-turn tasks, then REFUEL would be exactly the same as LT-MIXED, LT-ONLINE, and MT-MIXED as the length of the dialogue, $H$, is 1.
>
> The main reason that REFUEL’s performances on the previous two turns are worse than DPO and REBEL is due to the difference in how the reward is obtained. While DPO and REBEL optimize for the reward at the current turn, REFUEL regresses to a relative future reward. This difference means that models trained with REFUEL are designed to optimize for long-term outcomes, resulting in responses that prioritize future rewards. In contrast, models trained with DPO or REBEL tend to generate responses that maximize immediate reward. For shorter dialogues, such as those with only one or two turns, this short-term greedy approach leads to higher winrates for DPO and REBEL. However, this greedy strategy is less effective in longer dialogues, where long-term planning becomes more important.
>
> > Why are dialogues filtered to a maximum of 5 turns? It would seem that the performance improvement would be larger on longer dialogues.
>
> Thank you for this insightful suggestion. We agree that extending the dialogue length beyond 5 turns could potentially lead to improved results. However, we decided against this for two main reasons. First, the majority of dialogues in the Ultrainteract dataset are shorter than 5 turns, so comparing against baselines like LT-OFFLINE, LT-MIXED, and MT-MIXED would not be entirely fair on dialogues more than 5 turns, as these baselines do not include sufficiently long dialogues in their offline datasets. Second, generating dialogues longer than 5 turns would be computationally prohibitive for us, as REFUEL requires multiple iterations (iter 1 and iter 2), and we rely on a large user simulation model (Llama-3.1-70B-it). The computational demands for generating dialogues of that length exceed our current capacity.

---

### Official Review · Reviewer_iiTf · 2024-11-04

**Soundness:** 4
**Presentation:** 3
**Contribution:** 3
**Rating:** 6
**Confidence:** 2

**Summary:**

The manuscript introduces a training technique, grounded in RL theory/practice, for improving multi-turn dialogue generation using large-language models. The key problem addressed is that in standard multi-turn dialogue generation, training is done usually by relying on single-turn techniques, i.e. optimizing the last step of a dialogue conditioning on all previous steps. The manuscript calls out as this being a problem, since the policy generating previous data is not the one being optimized and refers to it as *covariate shift*. The proposed solution is grounded in Q-learning, avoiding the need for training separate actor / critic networks which are cumbersome to maintain and unstable to train.

The manuscript points out that with Transformers state can easily be restored (by just feeding back in a partial sequence) and alternative (counterfactual) steps can be generated (by just sampling). Then through a series of tricks the policy training can be tied to the change in relative rewards.

The evaluation setup leverages UltraInteract, making use of a LLama-8B for experiments and LLama-70B for simulating a user. Each turn quality (i.e. win-rate over other responses) is determined using GPT-4. The results clearly indicate improvements on turns 3, 4, 5 but are below baselines on turns 1, 2.

**Strengths:**

Some strengths include the following:

* The construction of the optimization algorithm I find novel and elegant.

* Very strongly grounded in theory of RL with the most being entirely explained from the common formulations of Q-learning paired with optimiziation techniques. Rather clearly written overall, as well taking into account less familiar readers with the details of the theory through the presence of a "Intuitive explanation" section.

* The use of two benchmarks (UltraInteract, AnthropicHH), as well as open-source models (LLama, ArnoRM) make results reproducible and ground the contributions.

**Weaknesses:**

Although I could follow the theory behind the paper and I'm familiar with most of it, I'm not the best reviewer for the correctness of the derivation of the formulations and final objective trained. Therefore my confidence is rather low on this aspect.

Some weaknesses I'd like to ask authors about:
* Why not extend beyond H=5? This seems rather limiting and although improvements are noticeable in the GPT-4 evaluations, presumably the benefits should be much stronger at say H>10.

* Although it is commonly accepted practice for GPT-4 to evaluate win-rates (or quality as judge), are there other measurements that can help quantify usefulness of REFUEL? For example, the qualitative examples in Appendix D. indicate repetitive text by other baselines. Yet looking at green highlights of long responses for math problems, seems to indicate some reasoning improvements.

* Can the authors comment on utility beyond multi-turn dialogue generation, e.g. perhaps for multi-step reasoning or tool-use?

**Questions:**

See weaknesses discussion.

---

> ### Author Response · Authors · 2024-11-19
> **Response to Reviewer iiTf**
>
> Thank you for your constructive feedback. We address each of your points below.
>
> > Why not extend beyond H=5? This seems rather limiting and although improvements are noticeable in the GPT-4 evaluations, presumably the benefits should be much stronger at say H>10.
>
> We appreciate this valuable suggestion and agree that extending beyond 5 turns could potentially yield even better results. However, we chose not to do so for two key reasons. First, the majority of dialogues in the Ultrainteract dataset consist of fewer than 5 turns. Comparing against baselines such as LT-OFFLINE, LT-MIXED, and MT-MIXED would therefore not be fair on dialogues longer than 5 turns, as these baselines do not contain sufficiently long dialogues in the offline dataset. Second, generating dialogues with more than 5 turns would be infeasible for our compute resources during data generation, as REFUEL requires multiple iterations (both Iter 1 and Iter 2), and we rely on a large user simulation model (Llama-3.1-70B-it). The computational cost for generating dialogues of such length is prohibitive at our current scale.
>
> > Although it is commonly accepted practice for GPT-4 to evaluate win-rates (or quality as judge), are there other measurements that can help quantify usefulness of REFUEL? For example, the qualitative examples in Appendix D. indicate repetitive text by other baselines. Yet looking at green highlights of long responses for math problems, seems to indicate some reasoning improvements.
>
> > Can the authors comment on utility beyond multi-turn dialogue generation, e.g. perhaps for multi-step reasoning or tool-use?
>
> Since both of these questions address the potential applications of REFUEL in reasoning tasks, we will answer them together.
>
> Multi-step reasoning can be viewed as a simplified version of multi-turn dialogue, where the model autonomously performs multiple reasoning steps after an initial user query. In contrast to multi-turn dialogue, where the trajectory consists of alternating turns between the human and assistant (e.g., $x_1, y_1, x_2, …, y_{h-1}, x_h, y_h$), multi-step reasoning focuses on a single human query followed by multiple reasoning steps from the model (e.g., $x_1, y_1, y_2, …, y_{h-1}, y_h$). REFUEL could similarly be applied to this task, where the state $s_h$​ would consist of the initial query $x_1$​ and all reasoning steps up to $h$: $s_h = (x_1, y_1, …, y_{h-1})$.
>
> For this paper, we focus on dialogue generation and thus use GPT-4 winrate, reward model scores, and KL-divergence as our primary metrics. However, we believe that a promising direction for future work is to apply REFUEL to enhance reasoning capabilities. In such a scenario, various evaluation methods could be used and the simplest one could be directly comparing the final answer similar to GSM8K [1], resulting in a binary reward for correctness.
>
> [1] Cobbe, Karl, et al. "Training verifiers to solve math word problems." arXiv preprint arXiv:2110.14168 (2021).

---

> ### Comment · Reviewer_iiTf · 2024-11-24
> **Thank you for your response**
>
> Thank your for the response. I believe the responses addresses my questions to some degree. While I definitely agree with the *potential* impact of REFUEL, some actual experimental evidence to confirm the claims would be needed to mandate a higher score. This could mean results on longer dialogue tasks or such multi-step reasoning problems discussed in the answer by the authors. Therefore, I will keep my score.

---

### Official Review · Reviewer_Lbbb · 2024-11-05

**Soundness:** 3
**Presentation:** 3
**Contribution:** 4
**Rating:** 6
**Confidence:** 4

**Summary:**

Long-term and multi-turn planning remains a recognized limitation in large language models (LLMs), limiting their effectiveness in complex, goal-oriented scenarios such as extended dialogue interactions. While reinforcement learning from human feedback (RLHF) with extended context lengths has been employed to address this, these methods encounter high computational costs and are vulnerable to covariate shift. This work presents REFUEL, a novel reinforcement learning framework designed for long-term planning. REFUEL optimises the model for the relative future, thus mitigating covariate shift.

**Strengths:**

- Presents an interesting contribution to the area of RL in LLM's, making a stride in solving the problem of longer term planning.
- Illustrates the effectiveness of the technique not only on large models but also on smaller LLMs.
- Presents and proves performance guarantees of the approach.

**Weaknesses:**

- REFUEL was only applied to the Llama 3 family of LLMs and further only the instruction tuned versions of these models. It would be more complete if the approach was also tested on models with different pretraining, e.g. the supervised language modelling pretrained models, models such as Gemma or Phi etc.
-  The datasets used for evaluation involve reasoning and harmful and helpful conversations with LLMs, but true goal oriented datasets such as those in task-oriented dialogue would present more of a long term planning challenge to test this approach.

**Questions:**

- Could this approach also be applied to LLMs only trained with the language modelling objective?
- How would this approach perform in more challenging planning scenarios such as task-oriented dialogue?

---

> ### Author Response · Authors · 2024-11-19
> **Response to Reviewer Lbbb**
>
> Thank you for your valuable review of our paper. We respond to your individual questions below.
>
> > REFUEL was only applied to the Llama 3 family of LLMs and further only the instruction tuned versions of these models. It would be more complete if the approach was also tested on models with different pretraining, e.g. the supervised language modelling pretrained models, models such as Gemma or Phi etc.
>
> > Could this approach also be applied to LLMs only trained with the language modelling objective?
>
> The Llama 3 family represents the current state-of-the-art in open-source large language models. By demonstrating that REFUEL improves performance on these models, we establish a strong foundation for its generalizability. Models like Gemma, which exhibit comparable performance to Llama 3 models [1], are likely to benefit similarly from REFUEL.
>
> For models trained only with the language modeling objective, instruction fine-tuning (IFT) would likely be needed before applying REFUEL. This is because REFUEL relies on datasets generated by the current policy $\pi_t$ (step 3 in Algorithm 1 of our paper). If a model is not good at following instructions, the datasets it generates will be poor in quality. As a result, much of the early training effort would go toward improving the model’s ability to follow instructions rather than focusing on long-term planning. It is important to note that this requirement is not unique to REFUEL. In general, an IFT step is typically needed before RLHF for training general-purpose chatbots [5, 6].
>
> > The datasets used for evaluation involve reasoning and harmful and helpful conversations with LLMs, but true goal oriented datasets such as those in task-oriented dialogue would present more of a long term planning challenge to test this approach.
>
> > How would this approach perform in more challenging planning scenarios such as task-oriented dialogue?
>
> While we did not explicitly evaluate REFUEL in the task-oriented dialogue domain, we suspect that it would also perform well in such scenarios. Task-oriented dialogues typically involve structured and predictable interactions, with lower variance in user responses compared to open-ended reasoning tasks [2, 3, 4]. This reduced variance in user behavior would lead to lower variance in unbiased Q-value estimates, thereby simplifying optimization and making the training process more stable.
>
> In contrast, reasoning tasks demand abstract and open-ended interactions, posing greater challenges for long-term planning due to their high variance. REFUEL’s strong performance in these complex tasks suggests it is well-suited to the structured, goal-driven nature of task-oriented dialogues. Testing REFUEL on such datasets is an exciting direction for future work, and we appreciate the suggestion.
>
> [1] Team, Gemma, et al. "Gemma 2: Improving open language models at a practical size." arXiv preprint arXiv:2408.00118 (2024).
>
> [2] Hosseini-Asl, Ehsan, et al. "A simple language model for task-oriented dialogue." Advances in Neural Information Processing Systems 33 (2020): 20179-20191.
>
> [3] Budzianowski, Paweł, et al. "Multiwoz--a large-scale multi-domain wizard-of-oz dataset for task-oriented dialogue modelling." arXiv preprint arXiv:1810.00278 (2018).
>
> [4] Eric, Mihail, and Christopher D. Manning. "Key-value retrieval networks for task-oriented dialogue." arXiv preprint arXiv:1705.05414 (2017).
>
> [5] OpenAI. (2023). GPT-4 technical report.
>
> [6] Dubey, A., Jauhri, A., Pandey, A., Kadian, A., Al-Dahle, A., Letman, A., ... & Ganapathy, R. (2024). The llama 3 herd of models. arXiv preprint arXiv:2407.21783.

---

### Meta-Review · Area_Chair_bh2Y · 2024-12-18

**Metareview:**

The authors address the problem that LLMs have with multi-turn tasks (such as dialogue) due to covariate shift when using standard RLHF approaches. They propose a framework called REFUEL that addresses this by framing multi-turn RLHF as a series of regression tasks on self-generated data, achieving superior performance to existing methods and even enabling a smaller model to outperform a much larger one in long dialogues. The reviewers appreciate the novelty and theoretical grounding, the empirical results, and the clarity of presentation. However, they also raised some significant concerns primarily on the limited scope of evaluation (model and task diversity, evaluation metrics, ablations), performance degradations not adequately explained, and lack of comparison to PPO. The authors respond to the comments but the reviewers are not convinced.

**Additional Comments On Reviewer Discussion:**

The authors attempt to address the comments and justify their choices (e.g. which models to evaluate, number of turns, alternative metrics etc.) but only one reviewer responds and they are not convinced to increase their score. Looking at the author responses, however, it feels like this work is very promising.

---

### Decision · Program_Chairs · 2025-01-22

Accept (Poster)